# Fairness in Link Prediction Beyond Demographic Parity: A Reproducibility Study

**Valentijn Oldenburg**                                   *valentijn.oldenburg@student.uva.nl*
*University of Amsterdam*

**Floris de Kam**                                         *floris.de.kam@student.uva.nl*
*University of Amsterdam*

**Stef de Wildt**                                         *stef.de.wildt@student.uva.nl*
*University of Amsterdam*

**Jarno Balk**                                            *jarno.balk@student.uva.nl*
*University of Amsterdam*

**Reviewed on OpenReview:** *https://openreview.net/forum?id=QNCZoPb9uV*

## Abstract

In fair ranked link prediction, demographic parity ($\Delta_{\mathrm{DP}}$) is a common fairness metric. Yet, Mattos et al. (2025) argue that it fails to detect exposure bias because it ignores where links appear in the ranking. In this study, we reproduce this claim by showing that $\Delta_{\mathrm{DP}}$ can indicate aggregate parity even when some subgroup-pair links are systematically ranked lower than others. The proposed rank-aware Normalized Discounted KL-divergence (NDKL), however, does detect such disparities. We also reproduce the effectiveness of MORAL, a post-processing method that improves exposure-based fairness while maintaining competitive utility. Beyond reproduction, we assess robustness using synthetic homophily settings, categorical sensitive attributes, and additional fairness and utility metrics, including subgroup-pair-adapted Attention-Weighted Rank Fairness (AWRF). Overall, our results show that exposure-based metrics uncover biases hidden by $\Delta_{\mathrm{DP}}$ and that MORAL reduces these biases with minimal utility loss across diverse settings and datasets. We release a corrected, reproducible implementation at `https://github.com/Floris93100/reproducing-MORAL`.

## 1 Introduction

Fairness in link prediction aims to reduce disparities in predicted link probabilities across groups (Chen et al., 2024). In social-network recommendations, link predictors shape who becomes visible and connected (Ferrara et al., 2022) and can induce feedback loops that reinforce within-group connections and disadvantage minority groups (Karimi et al., 2018). Importantly, even small biases can have large downstream impacts on minority groups (Singh & Joachims, 2018). For example, in homelessness support systems, data-driven tools are used to distribute scarce services (Wilde et al., 2021; Moon et al., 2025), and if clients with certain sensitive attributes are ranked lower, they may receive fewer referrals or slower access to support (Rice & Young, 2025). This is especially concerning because link predictors are trained on historical graph data and can therefore inherit and amplify existing structural biases (Dai et al., 2024).

At the model level, such amplification can lead to prediction distortions, including exposure skews in ranked outputs (Gupta et al., 2021), degree-driven popularity effects (Subramonian et al., 2023), and hidden subgroup disparities (Kearns et al., 2018). Mitigating these biases is challenging because candidate links are structurally dependent (non-i.i.d.) and subject to temporal evolution and feedback (Stoica et al., 2018; Zhang et al., 2024). Also, limited expressivity of graph neural nets (GNNs) can make nodes with similar local graph structure but different sensitive attributes, such as sex or age, indistinguishable (Mattos et al., 2025). These concerns have motivated recent work on defining and enforcing fairness in link prediction (Chen et al., 2024).

In this spirit, Li et al. (2021) propose a *dyadic* definition of fairness that groups links into two coarse categories, same-group links and cross-group links, where fairness requires similar prediction rates for these two categories. In this dyadic view, demographic parity holds when the categories are predicted at similar rates. However, Mattos et al. (2025) argue that merging distinct subgroup-pair types into these two broad categories can hide disparities between specific pairs of subgroups, even when overall demographic parity holds. For example, male and female nodes may receive similar overall exposure, even though female-female links are ranked lower. In other words, dyadic demographic parity ignores where links appear in the ranking. This suggests that dyadic demographic parity, despite its use in recent literature (e.g., Current et al., 2022; Luo et al., 2023), may be too coarse for fair link prediction. To support these claims, Mattos et al. (2025) (*i*) formalise and empirically validate limitations of dyadic demographic parity, (*ii*) introduce the rank-aware, exposure-based *Normalized Discounted KL-divergence* (**NDKL**) metric, and (*iii*) propose **MORAL**, a post-processing algorithm that re-ranks outputs of decoupled link predictors in an exposure-aware way.

In this study, we reproduce these claims and release a corrected implementation. Beyond reproduction, we extend the original work through (*i*) an asymmetric homophily stress-test with synthetic graphs, (*ii*) a metric robustness analysis, and (*iii*) experiments with categorical sensitive attributes. Overall, our results support exposure-based, rank-aware fairness in link prediction by showing that dyadic demographic parity can obscure subgroup-pair exposure disparities and that MORAL reduces them across diverse settings.

## 2 Scope of reproducibility

The objective of this study is to reproduce and extend the key findings of Mattos et al. (2025).[1] Along with proposing the **MORAL** post-processing algorithm and re-introducing the **NDKL** metric, the paper makes three key claims, labeled **C1**, **C2**, and **C3** for reference throughout the text:

- **Claim 1 (C1): Demographic parity hides within-group exposure bias.** The authors reason and empirically show how aggregating across subgroup-pair types in dyadic fairness obscures *within-*group disparities, even while group-level fairness and demographic parity are satisfied.

- **Claim 2 (C2): NDKL overcomes the limitations of demographic parity.** Unlike demographic parity, NDKL is a non-dyadic, rank-aware metric. It explicitly treats all subgroup-pairings and detects exposure disparities that demographic parity fails to detect.

- **Claim 3 (C3): MORAL achieves strong fairness-utility trade-offs.** The paper shows how MORAL consistently mitigates previously undetected exposure biases, as measured by NDKL. Meanwhile, MORAL keeps its competitive utility relative to baselines, as measured by Precision@$K$.

To confirm these claims, we reproduce the core experimental setup and test it across all datasets using the proposed fairness and utility metrics. Moreover, to further assess robustness and generality, we extend the original evaluation along variations in homophily, evaluation metrics, and sensitive attribute cardinality:

- **Asymmetric homophily stress-test:** Increasing homophily progressively reduces certain subgroup-pair types, thereby stressing MORAL's ability to reallocate exposure. We evaluate how MORAL's fairness-utility trade-offs evolve as subgroup interactions become more constrained.

- **Metric robustness analysis:** Since MORAL is explicitly optimised w.r.t. NDKL, we additionally evaluate fairness using a metric that differs in how subgroup exposure is aggregated. For utility, we add a rank-aware measure that accounts for the positions of correct predictions.

- **Categorical sensitive attributes:** We evaluate whether the proposed framework generalises beyond binary sensitive attributes to categorical sensitive attributes, assessing the practical validity of the paper's claim that MORAL extends to higher-cardinality settings.

Specifically, the homophily stress-test primarily explores **C3**, the metric robustness analysis targets **C2** & **C3**, and the attribute cardinality extension tests whether **C1**–**C3** generalise beyond binary sensitive attributes.

---

[1]We reproduce the 2025 arXiv version, which was later published in shortened form at AAAI 2026 (Mattos et al., 2026).

# 3 Methodology

This section describes the methodology and experimental setup required to reproduce and evaluate the claims by Mattos et al. (2025). In Section 3.6, we describe the extensions that go beyond reproduction.

## 3.1 Preliminaries and ranking setup

We first formalise link prediction as a ranking problem, then introduce how sensitive attributes induce subgroup-pair types, and end by motivating the role of exposure in ranked outputs.

**Link prediction as ranking.** Let $\mathcal{G} = (\mathbb{V}, \mathbb{E}, \mathbf{s})$ be a graph with node set $\mathbb{V}$, edge set $\mathbb{E}$, and sensitive attribute vector $\mathbf{s} \in \{0,1\}^{|\mathbb{V}|}$ with $s_v$ as the sensitive attribute of node $v$. Let $\mathbb{C} \subseteq \mathbb{V} \times \mathbb{V}$ denote the set of candidate node pairs.[2] A link predictor assigns each pair $(u,v) \in \mathbb{C}$ a score $f(u,v) \in [0,1]$ (e.g., using a GNN). Sorting candidate pairs in decreasing score order yields a ranking $\mathbf{R} = (R_1, R_2, \dots)$ where $\mathbf{R}$ is an ordered set and $R_k \in \mathbb{C}$ denotes the pair at rank $k$. We evaluate the models on $\mathbf{R}$'s top-$K$ list.

**Sensitive attributes and subgroup-pair types.** The sensitive attribute divides candidate node pairs into *subgroup-pair types* based on the attributes of the two nodes involved. For binary sensitive attributes $s \in \{0,1\}$ with unordered pairs, this yields three pair types: $\mathbb{E}_{0\text{-}0} = \{(u,v) \in \mathbb{C} \mid s_u = 0, s_v = 0\}$, $\mathbb{E}_{1\text{-}1} = \{(u,v) \in \mathbb{C} \mid s_u = 1, s_v = 1\}$, and $\mathbb{E}_{0\text{-}1} = \{(u,v) \in \mathbb{C} \mid s_u \neq s_v\}$. Of these, $\mathbb{E}_{0\text{-}0}$ and $\mathbb{E}_{1\text{-}1}$ belong to the intra-group $\mathbb{E}_{\text{intra}} = \mathbb{E}_{0\text{-}0} \cup \mathbb{E}_{1\text{-}1}$, while $\mathbb{E}_{0\text{-}1}$ belongs to the inter-group $\mathbb{E}_{\text{inter}} = \mathbb{E}_{0\text{-}1}$. While dyadic fairness aggregates links to the coarse group-level of $\mathbb{E}_{\text{intra}}$ and $\mathbb{E}_{\text{inter}}$ (Li et al., 2021), Mattos et al. (2025) argue this can obscure finer-grained disparities happening *within* $\mathbb{E}_{\text{intra}}$ and $\mathbb{E}_{\text{inter}}$.

**Exposure in rankings.** Because link prediction outputs are often used downstream as ranked lists, fairness is inherently tied to exposure: pairs ranked higher receive disproportionately more attention (Abdollahpouri, 2019). To reflect this asymmetry in attention, we relate each rank $k$ with a logarithmically decaying factor

$$\delta_k = \frac{1}{\log_2(k+1)}, \tag{1}$$

such that deviations in predictions of higher ranks are penalised more heavily than those at lower ones.

## 3.2 Fairness and utility metrics

This section formalises the fairness and utility metrics adapted from Mattos et al. (2025): (*i*) dyadic demographic parity, (*ii*) the non-dyadic, rank-aware NDKL, and (*iii*) the utility metric.

**Demographic parity ($\Delta_{\text{DP}}$).** Dyadic demographic parity ($\Delta_{\text{DP}}$) compares link predictions $f(u,v) \in [0,1]$ of intra-group $\mathbb{E}_{\text{intra}}$ and inter-group $\mathbb{E}_{\text{inter}}$ (both defined in Section 3.1), giving

$$\Delta_{\text{DP}}@K = \left| \frac{\sum_{(u,v) \in \mathbb{R}_{1:K} \cap \mathbb{E}_{\text{intra}}} f(u,v)}{|\mathbb{R}_{1:K} \cap \mathbb{E}_{\text{intra}}|} - \frac{\sum_{(u,v) \in \mathbb{R}_{1:K} \cap \mathbb{E}_{\text{inter}}} f(u,v)}{|\mathbb{R}_{1:K} \cap \mathbb{E}_{\text{inter}}|} \right|, \tag{2}$$

where $\mathbb{R}_{1:K}$ denotes the set of candidate node pairs in the top-$K$ of $\mathbf{R}$.[3] Equivalently, $\Delta_{\text{DP}}$ requires independence between predicted links and sensitive attributes at the level of $\mathbb{E}_{\text{intra}}$ and $\mathbb{E}_{\text{inter}}$ (Gajane & Pechenizkiy, 2017). Although this dyadic aggregation into $\mathbb{E}_{\text{intra}}$ and $\mathbb{E}_{\text{inter}}$ is used in prior work (e.g., Li et al., 2021; Current et al., 2022; Li et al., 2022; Luo et al., 2023), it can obscure disparities within those groups. As a result, exposure imbalances can persist even when $\Delta_{\text{DP}}$ holds, thus giving a theoretical basis for Claim **C1**.

**Fairness metric: Normalised Discounted KL-divergence (NDKL).** To overcome $\Delta_{\text{DP}}$'s limitations, Mattos et al. (2025) propose two properties a metric for fair link prediction should have. Here, $\boldsymbol{\pi}$ denotes a discrete probability distribution over $T$ subgroup-pair types, and $\hat{\boldsymbol{\pi}}_k(\mathbf{R})$ the predicted distribution induced by the top-$k$ prefix of $\mathbf{R}$. In the binary case, this gives $\boldsymbol{\pi} = (\pi_{0\text{-}0}, \pi_{0\text{-}1}, \pi_{1\text{-}1})$ and $\hat{\boldsymbol{\pi}}_k = (\hat{\pi}_{k,0\text{-}0}, \hat{\pi}_{k,0\text{-}1}, \hat{\pi}_{k,1\text{-}1})$.

---

[2]Here we use blackboard bold (e.g., $\mathbb{C}$) to denote sets; this should, for example, not be mistaken for the complex set.

[3]Mattos et al. (2025) define $\Delta_{\text{DP}}$ in general form as $\Delta_{\text{DP}} = \left| p(\hat{Y} = 1 \mid (u,v) \in E_{\text{intra}}) - p(\hat{Y} = 1 \mid (u,v) \in E_{\text{inter}}) \right|$, where $p(\cdot)$ is taken over the distribution of $(u,v) \in \mathbb{C}$ and $\hat{Y} = 1$ denotes a positive prediction. Eq. 2 is our adaptation of this definition to the empirical top-$K$ ranking setting: it uses mean predicted link scores for $\mathbb{E}_{\text{intra}}$ and $\mathbb{E}_{\text{inter}}$ within $R_{1:K}$.

- **Property 1: Non-dyadic distribution-preserving fairness.** A fairness metric should treat all subgroup-pair types distinctly and aim to align the predicted distribution $\hat{\boldsymbol{\pi}}$ with the original $\boldsymbol{\pi}$.
- **Property 2: Rank-awareness.** A fairness metric should be sensitive to the proportion and exposure of every type of pair. Specifically, it should penalise deviations from $\boldsymbol{\pi}$ according to $\sum_{k=1}^{|\mathbb{C}|} \text{dist}(\hat{\boldsymbol{\pi}}_k(\mathbf{R}), \boldsymbol{\pi})\delta_k$, where $\text{dist}(\cdot, \cdot)$ is some distance function and $\delta_k$ enforces rank-awareness.

Following these properties, the NDKL (first introduced by Geyik et al. (2019)) is defined as

$$\text{NDKL@}K = \frac{1}{Z}\sum_{k=1}^{K}\delta_k D_{\text{KL}}(\hat{\boldsymbol{\pi}}_k, \boldsymbol{\pi}), \tag{3}$$

where the distance function is the KL-divergence and $Z = \sum_{i=1}^{K}\frac{1}{\log_2(i+1)}$ is a normaliser. Therefore, the NDKL satisfies Properties 1 & 2, consistent with the theoretical motivation for Claim **C2**.

**Utility metric: Precision@1000.** We adopt Precision@$K$, the fraction of true edges among the top-$K$ candidate pairs, using $K = 1000$. Unlike the NDKL, it is insensitive to subgroup composition and exposure.

### 3.3 MORAL post-processing framework

This section describes the framework behind *Multi-Output Ranking Aggregation for Link Fairness* (MORAL). MORAL aims to overcome three deficiencies of dyadic fairness in link prediction (Mattos et al., 2025): ($i$) limited expressivity[4] of standard (non-decoupled; single-model) message-passing GNNs that can blur subgroup-pair-specific asymmetries, ($ii$) subgroup-pair imbalances hidden by aggregation of $\mathbb{E}_{\text{intra}}$ or $\mathbb{E}_{\text{inter}}$ under $\Delta_{\text{DP}}$, and ($iii$) $\Delta_{\text{DP}}$'s permutation invariance to ranking order that ignores exposure.

**Decoupled subgroup-pair predictors.** MORAL requires decoupled link predictors per subgroup-pair type; in the binary setting of Section 3.1, that means models $f_{\text{0-0}}, f_{\text{1-1}}, f_{\text{0-1}}$ for $\mathbb{E}_{\text{0-0}}, \mathbb{E}_{\text{1-1}}$, and $\mathbb{E}_{\text{0-1}}$, respectively, where each model is trained solely on edges of its type and outputs logits for these edges. Following training, the candidate pairs of each subgroup-pair type $(u_i, v_i) \in \mathbb{C} \cap \mathbb{E}_j$, where $j \in \{\text{0-0, 1-1, 0-1}\}$, are sorted using the logits into a decreasingly ordered set $\mathbf{C}_j = ((u_1, v_1, \text{score}_1), (u_2, v_2, \text{score}_2), \ldots)$.

**Greedy KL-guided aggregation.** MORAL's objective is to, given the ordered sets $\mathbf{C}_{\text{0-0}}, \mathbf{C}_{\text{1-1}}, \mathbf{C}_{\text{0-1}}$ and a target exposure distribution $\boldsymbol{\pi}$ over subgroup-pair types, construct a ranking $\mathbf{R}$. $\boldsymbol{\pi}$ is set to the observed proportions in the original graph (also reflecting **Property 1**, used by the NDKL). At each rank position, MORAL selects the current top remaining edge from each $\mathbf{C}$ and chooses the one that minimises $D_{\text{KL}}(\boldsymbol{q}' \| \boldsymbol{\pi})$, where $\boldsymbol{q}'$ is the updated distribution if that subgroup-pair type were chosen (see Appendix A for the full algorithm). This greedy KL-divergence-minimising procedure mirrors the subgroup-aware, exposure-based mechanism of the NDKL (Eq. 3) — a relationship we further detail in Section 3.6.2. (To isolate the effect of this reranking relative to the decoupled predictors, Appendix G reports a pre- vs. post-rerank ablation.)

Notably, $\boldsymbol{\pi}$ is a design choice that, by choosing distribution-preserving fairness, aims to prevent algorithmic bias, rather than it being a universally correct definition of fairness (Mitchell et al., 2021). In this sense, because link predictors can inherit and amplify structural bias in ranked outputs (Gupta et al., 2021; Dai et al., 2024), Property 1 and MORAL's distribution-preserving target should be interpreted as a non-amplification objective, not a corrective one. In other words, if the original $\boldsymbol{\pi}$ reflects historical bias, then low NDKL just indicates agreement of the predicted $\hat{\boldsymbol{\pi}}$ with the original $\boldsymbol{\pi}$, not with fairness in a broader normative sense.

### 3.4 Datasets

Six graph datasets on social networks are used, detailed in Table 1.[5] Pokec-n and Pokec-z are samples from Pokec (Takac & Zabovsky, 2012), a Slovakian social network. The datasets' topologies are of type *Community* and *Periphery*. (Appendix F discusses MORAL's performance on datasets with a different structure/mixing pattern.) Fairness concerns are particularly relevant for social networks, where factors such as historical inequality and homophily can be amplified algorithmically (Stoica et al., 2018; Barocas et al., 2023).

---

[4]In particular, the authors note that standard message-passing GNNs are at most as expressive as 1-WL (Xu et al., 2018)

[5]The datasets are retrieved from: `https://github.com/yushundong/PyGDebias` (Dong et al., 2023).

Table 1: Social graph datasets used by Mattos et al. (2025). More details are listed in Appendix C.1.

| Dataset | Network type | Node | Edge | Sensitive attribute (2-ary) |
|---------|--------------|------|------|------------------------------|
| CREDIT | Credit network | Individual | Credit relationship | Age (binned) |
| FACEBOOK | Online medium | User | Friendship | Gender (male vs. female) |
| GERMAN | Credit approval | Individual | Similarity link | Gender (male vs. female) |
| NBA | Athletes on Twitter | Athlete | Twitter connection | Nationality (US vs. non-US) |
| POKEC-N | Online medium | User | Friendship | Gender (male vs. female) |
| POKEC-Z | Online medium | User | Friendship | Gender (male vs. female) |

### 3.5 Experimental setup and code

This section describes the experimental pipeline for reproducing the experiments by Mattos et al. (2025). While the authors provide an open-source repository[6] and report full out-of-the-box reproducibility, we find the codebase omits several elements required for reproduciblity and is inconsistent with the paper's formal definitions. Below, we summarise components that are directly reused, describe necessary corrections and extensions, list reimplemented baselines, and conclude with reproducibility details. Together, these corrections align the implementation with the paper's definitions for reliable evaluation of Claims **C1**–**C3**.

**Reused training pipeline.** We adopt the authors' implementation for training the decoupled predictors used by MORAL. Out of the box, the repository provides dataset loaders for all datasets in Table 1 and an extensive model class for training multiple graph neural network architectures (Gori et al., 2005; Wu et al., 2020), including training and evaluation routines (using binary cross-entropy), and the option for different kinds of encoder and decoder models. Following Mattos et al. (2025), we use a GCN (Kipf & Welling, 2016) encoder and a dot-product decoder, where the decoder defines the scoring function $f$.

**Inconsistencies.** We identify several discrepancies between the implementation and the paper's formal definitions. Notably, the target distribution is computed from node labels rather than subgroup-pair proportions, giving a fundamentally different distribution. Also, demographic parity is computed by conditioning on the sensitive attribute of one node of an edge, rather than on edge membership in $\mathbb{E}_{\text{intra}}$ and $\mathbb{E}_{\text{inter}}$.

For the Pokec datasets, the sensitive attributes alternates between `Gender` and `Region` between the paper text, paper tables, and code; we follow the paper text and adopt `Gender` (as listed in Table 1). Finally, MORAL's greedy KL-aggregation step (introduced in Section 3.3) differed from the pseudocode (specified in Appendix A.1): it operates on unnormalised group counts rather than probability distributions, which, for instance, breaks scale-independence and corrupts the KL-divergence computation.

**Additional components required for reproduction.** Two components required for reproduction are missing: the NDKL and a data-splitting procedure. After correspondence with the authors, we received an NDKL implementation. Since it differs slighty from the formal definition in its normalization and divergence computation, we reimplement NDKL exactly as specified in Section 3.2 with all details in Appendix A.2.

For data splits, the paper specifies 70/20/10% train/validation/test proportions but omits the negative sampling scheme; hence, we implement a stratified split protocol matching these proportions. Importantly, all graphs are treated as undirected by removing self-loops, merging duplicate edges, and treating $(u, v)$ as equivalent to $(v, u)$. Positive edges are grouped by subgroup-pair type and randomly assigned to splits; negative edges are sampled independently per split to ensure (*i*) no negative edge coincides with a positive one, (*ii*) no duplicates occur, and (*iii*) subgroup-pair proportions match those of positive edges.

**Baselines.** To put MORAL in context of prior work, we evaluate three baselines from Mattos et al. (2025): two in-processing methods and one post-processing re-ranker. For in-processing, we include (*i*) *FairAdj* (Li et al., 2021), which optimises link prediction under dyadic fairness constraints; and (*ii*) *FairWalk* (Rahman et al., 2019), a fairness-aware embedding method. For post-processing, we include (*iii*) *DetConstSort* (Geyik

---

[6]Mattos et al. (2025)'s GitHub code repository: `https://github.com/joaopedromattos/MORAL`.

et al., 2019), a deterministic reranker with prefix constraints. As baseline hyperparameters are not fully specified, we use released implementations where possible and report configurations in Appendix B.3.

**Reproducibility details.** Appendix B summarises the hyperparameters used for all experiments and models. To ensure reproducibility, we average over three random seeds that fix all sources of randomness. We document the Python environments and library versions in our GitHub repository.[7] All computations are executed on NVIDIA A100 Tensor Core GPUs of the Dutch national supercomputer, Snellius. The total computational expense for reproduction equals 17.96 GPU hours with an estimated 1.24 kgCO$_2$eq.[8]

### 3.6 Extensions beyond original work

The following subsections provide an explanation of the extensions (first introduced in Section 2) to the foundational work by Mattos et al. (2025). These extensions test whether the original claims remain meaningful under harder and more realistic conditions along three complementary dimensions. The homophily stress-test examines whether MORAL still works when the graph structure makes some subgroup-pair links scarce, which is when exposure redistribution should be most difficult. The metric robustness analysis asks whether the reported gains reflect broader reductions in exposure disparity, rather than improvements tied only to the particular way NDKL aggregates exposure. The categorical-attribute extension tests whether the framework remains useful beyond the binary setting, which matters both for realism and for revealing how fairness-aware reranking scales as the number of subgroup pairs grows.

#### 3.6.1 Asymmetric homophily stress-test

Homophily is the tendency to connect to others sharing similar characteristics, such as sensitive group membership. In many types of networks, homophily co-occurs with preferential (i.e. degree-based) attachment, reinforcing within-group connectivity (Subramonian et al., 2023). This, in turn, affects the visibility of minority groups in degree-based rankings and recommender systems (Fabbri et al., 2020) and can exacerbate inequalities, such as the glass ceiling effect and invisibility syndrome (Avin et al., 2015; Karimi et al., 2018).

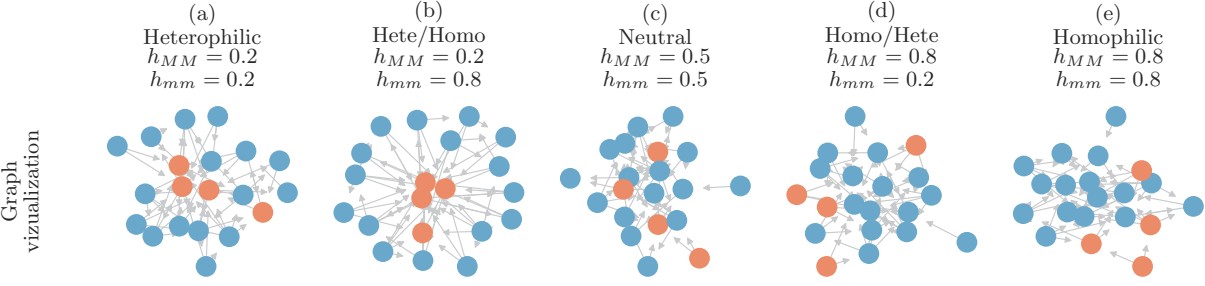

Figure 1: Examples of different homophily levels in graphs. Adapted from Espín-Noboa et al. (2022).

At high homophily, where homophily is quantified as the fraction of within-group edges $h_{\text{edge}}$

$$h_{\text{edge}}(\mathcal{G}) = \frac{1}{|\mathbb{E}|} \sum_{(u,v) \in \mathbb{E}} \mathbf{1}[s_u = s_v], \tag{4}$$

some subgroup-pair candidate pools become scarce, reducing training signal for decoupled predictors. To study the effect of asymmetric homophily on MORAL's fairness and utility, we generate synthetic graphs using the *Directed Preferential Attachment with Homophily* (DPAH) model by Espín-Noboa et al. (2022). DPAH combines preferential attachment and group-conditional homophily, where parameters $h_{\text{MM}}$ and $h_{\text{mm}}$ control the within-group attachment probability for newly arriving majority and minority nodes, respectively.

---

[7]Our implementation is available at: `https://github.com/Floris93100/reproducing-MORAL`.
[8]Emissions were estimated using CodeCarbon (Lacoste et al., 2023); the estimated carbon intensity of the Netherlands was 411 g CO$_2$eq kWh$^{-1}$ in January 2026 (Nowtricity, 2026). Appendix C.2 gives more details on this study's environmental impact.

For implementation, we adapt the DPAH model by Espín-Noboa et al. (2022) to include a binary sensitive attribute $s \in \{0, 1\}$, where $s = 0$ and $s = 1$ denote majority and minority nodes, respectively, following the setup of Figge et al. (2025). Node features are sampled i.i.d. as $\mathbf{x}_u \sim \mathcal{N}(\mathbf{0}, \mathbf{I}_d)$, such that features do not encode group membership. Figure 1 shows examples of different homophily levels by varying $h_{\mathrm{MM}}$ and $h_{\mathrm{mm}}$.

For our experiment, we keep the number of nodes and edge density fixed, while varying $h_{\mathrm{MM}}$, $h_{\mathrm{mm}}$, and minority fraction $f_{\mathrm{m}}$, such that we cover a broad range of edge distributions $\boldsymbol{\pi}$ (see Appendix B.2 for a hyperparameter overview). We vary $h_{\mathrm{MM}}$ and $h_{\mathrm{mm}}$ independently to study asymmetric homophily, since similar overall homophily can arise from different majority-minority mixing patterns in the graph structure. We expect higher homophily and smaller minority fractions to make some subgroup-pair candidate pools smaller, leading to exposure disparities detected by NDKL but missed by $\Delta_{\mathrm{DP}}$, since $\Delta_{\mathrm{DP}}$ ignores ranking position (Singh & Joachims, 2018). Across homophily settings, we keep the training and evaluation pipeline fixed and equal to our other experiments, isolating the effect of mixing.[9]

### 3.6.2 Metric robustness analysis

MORAL's greedy KL-guided aggregation selects edges to minimize divergence to the target distribution $\boldsymbol{\pi}$ at each rank position $k$ of $\mathbf{R}$. NDKL, in turn, evaluates $\mathbf{R}$ by accumulating KL-divergence to the same $\boldsymbol{\pi}$ over prefixes (Eq. 3). Because MORAL's ranking procedure directly optimizes the same divergence that NDKL evaluates post hoc (see Algorithm 1), MORAL's performance under NDKL may be particularly favourable.

To disentangle this coupling and assess the robustness of MORAL's reported fairness-utility trade-offs (Claim **C3**) beyond NDKL, we evaluate fairness with a non-dyadic, rank-aware metric that aggregates exposure differently. In addition, we inspect utility with a position-sensitive metric to complement the Precision@1000.

**Fairness metric: AWRF.** As described in Section 3.2, Mattos et al. (2025) argue a metric for fairness in link prediction should have **Property 1** (treating subgroup-pair types distinctly) and **Property 2** (rank-awareness). To adhere to these properties while also substituing the aggregation mechanism, we adopt *Attention-Weighted Rank Fairness* (AWRF) as a complementary fairness metric (Sapiezynski et al., 2019; Raj & Ekstrand, 2022). Whereas NDKL accumulates KL-divergence across all ranking pre-fixes, AWRF instead measures exposure in a single, position-weighted distribution over subgroup-pair types. As a result, both satisfy the properties by Mattos et al. (2025), yet differ fundamentally in how exposure is aggregated.

To formally adapt Sapiezynski et al. (2019)'s general AWRF definition to our task-specific purposes, we first define normalized position weights

$$\alpha_k = \frac{\delta_k}{\sum_{i=1}^{K} \delta_i}, \tag{5}$$

where $\delta_k$ and $\delta_i$ follow Eq. 1. Let $g : \mathbb{C} \to \{\text{0-0, 0-1, 1-1}\}$ map each candidate pair to its subgroup-pair type. The attention-weighted exposure distribution induced by $\mathbf{R}$ is then (by notation of Raj & Ekstrand (2022))

$$\epsilon(\mathbf{R})_j = \sum_{k=1}^{K} \alpha_k \, \mathbf{1}[g(R_k) = j], \tag{6}$$

where $\boldsymbol{\epsilon}$ is a probability distribution over subgroup-pair types with $\sum_j \epsilon(\mathbf{R})_j = 1$ and where $\epsilon(\mathbf{R})_j$ denotes component $j \in \{\text{0-0, 0-1, 1-1}\}$ of the distribution. Thus, for every $j$, $\epsilon(\mathbf{R})_j$ represents the total normalized exposure assigned to subgroup-pair type $j$. Given the target distribution $\boldsymbol{\pi}$, AWRF is then computed as

$$\mathrm{AWRF}(\mathbf{R}) = \mathrm{dist}(\boldsymbol{\epsilon}(\mathbf{R}), \, \boldsymbol{\pi}), \tag{7}$$

where, as distance function $\mathrm{dist}(\cdot, \cdot)$ we choose the $\ell_1$-norm to vary from the NDKL's KL-divergence and to ensure numerical stability, e.g., in cases where subgroup-pair types receive (near-)zero exposure.

In this definition, we adapt AWRF (Sapiezynski et al., 2019) in three ways: (*i*) we use one-hot subgroup-pair membership, rather than allowing probabilistic group membership; (*ii*) we fix the position weights $\alpha_k$ using Eq. 1's log-discount, rather than using an attention model; and (*iii*) we compare exposure to distribution-preserving target distribution $\boldsymbol{\pi}$, rather than using a population estimator (Raj & Ekstrand, 2022).

---

[9]The total computational expense for this extension equals 30.41 GPU hours with an estimated 1.41 kgCO$_2$eq.

**Utility metric: NDCG@1000.** To complement Precision@1000, we additionally report the *Normalized Discounted Cumulative Gain* (NDCG) at $K = 1000$ (Järvelin & Kekäläinen, 2002). Unlike Precision@$K$, NDCG@$K$ is sensitive to *where* predictions occur. Differences in their scores thus allow us to interpret (to some degree) which part of the ranking is predicted well. Formally, the discounted cumulative gain is

$$\text{DCG@}K = \sum_{k=1}^{K} \delta_k \mathbf{1}[\mathbf{R}_k \in \mathbb{E}_{\text{true}}], \tag{8}$$

where $\mathbb{E}_{\text{true}}$ is the set of true edges and $\delta_k$ follows Eq. 1. In turn, NDCG@$K$ = DCG@$K$/IDCG@$K$, where IDCG@$K$ is the DCG of an *Ideal* ranking s.t. the range is normalized to the unit interval (Manning, 2008).

### 3.6.3 Categorical sensitive attributes

In Section 3.1, we described how *binary* sensitive attributes $s \in \{0, 1\}$ partition candidate node pairs into subgroup-pair types. While Mattos et al. (2025) state this framework can be generalized in theory, they do not test this. Thus, to study their claim in practice, we extend MORAL to *categorical* sensitive attributes $s \in \mathbb{S} = \{0, 1, \ldots, m-1\}$ with cardinality $|\mathbb{S}| = m > 2$. We test the updated implementation up to $m = 10$.

Formally, we define the subgroup-pair type $\mathbb{E}_{a\text{-}b} = \{(u, v) \in \mathbb{C} \mid \{s_u, s_v\} = \{a, b\}\}$, where $0 \leq a \leq b \leq m-1$, yielding $T(m) = \frac{m(m+1)}{2}$ subgroup-pair types (the unique entries of a symmetric $m \times m$ matrix). We further define the intra- and inter-sets as $\mathbb{E}_{\text{intra}} = \{(u, v) \in \mathbb{C} \mid s_u = s_v\}$ and $\mathbb{E}_{\text{inter}} = \{(u, v) \in \mathbb{C} \mid s_u \neq s_v\}$. In the binary case, within-group disparites can only arise in the intra-group (since for $m = 2$, $|\mathbb{E}_{\text{intra}}| = 2$ and $|\mathbb{E}_{\text{inter}}| = 1$). In contrast, in the $m > 2$ categorical setting, these disparities can arise in both $\mathbb{E}_{\text{intra}}$ and $\mathbb{E}_{\text{inter}}$ (since for, e.g., $m = 3$, $|\mathbb{E}_{\text{intra}}| = |\mathbb{E}_{\text{inter}}| = 3$). More generally, with increasing $m$, aggregation collapses an increasing number of subgroup-pair interactions into $\mathbb{E}_{\text{intra}}$ and $\mathbb{E}_{\text{inter}}$, motivating a granular evaluation across all $T$ types with MORAL and NDKL, rather than at the coarse, aggregate level with $\Delta_{\text{DP}}$.

**Experimental setup.** We test the categorical extension on the Credit dataset by binning the `Age` attribute[10] into $m$ discrete categories. To accomodate any $m$ algorithmically, we modify MORAL to handle the $T$ induced subgroup-pair types dynamically: one decoupled encoder-decoder model is trained per type, and greedy KL-aggregation yields a $T$-dimensional ranking $\mathbf{R}$. For evaluation, both original and new metrics are adapted; $\Delta_{\text{DP}}$ (Eq. 2) is generalised using a One-vs-All approach, inspired by Figge et al. (2025).

## 4 Results

This section begins by presenting findings on the reproducibility of Claims **C1** through **C3**, introduced in Section 2. Following, extensions that go beyond the original paper are discussed.

### 4.1 Reproducibility of core claims

**Claim C1 & C2.** Claim **C1** states that $\Delta_{\text{DP}}$ hides within-group exposure bias while **C2** states that NDKL explicitly treats all subgroup-pairings and detects exposure disparities that $\Delta_{\text{DP}}$ fails to detect.

Figure 2 depicts $\Delta_{\text{DP}}$ and NDKL across a range of experimental configurations with binary sensitive attributes, such that exposure disparities can only arise within $\mathbb{E}_{\text{intra}}$. The configurations are grouped into bins by the log-ratio $\hat{\pi}_{0\text{-}0}/\hat{\pi}_{1\text{-}1}$, thus reflecting variation in exposure allocation within $\mathbb{E}_{\text{intra}} = \mathbb{E}_{0\text{-}0} \cup \mathbb{E}_{1\text{-}1}$. The ratio measures how exposure is redistributed between two edge types while their combined exposure is fixed. While $\Delta_{\text{DP}}$ remains largely invariant across bins, NDKL varies substantially, revealing its sensitivity to exposure shifts. Since only the relative allocation $\hat{\pi}_{0\text{-}0}/\hat{\pi}_{1\text{-}1}$ is changed, NDKL is maximized where this ratio is, on average, most misaligned with the baseline ratio $\pi_{0\text{-}0}/\pi_{1\text{-}1}$ implied by $\boldsymbol{\pi}$. This seems to occur around $\hat{\pi}_{0\text{-}0}/\hat{\pi}_{1\text{-}1} \approx 1$ in Figure 2. Consistent with **C1** and **C2**, the contrast between the two empirically confirms that $\Delta_{\text{DP}}$ obscures within-group exposure disparities (**C1**), whereas NDKL does capture them (**C2**).

---

[10] Here, we use the non-preprocessed version of the Credit dataset by Lichman (2013), downloadable from Hugging Face. We construct the adjacency matrix by connecting each node to its 10 nearest neighbours in the standardised feature space.

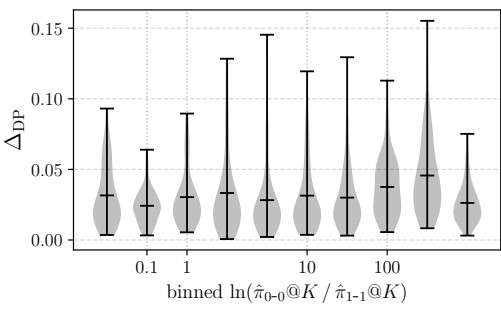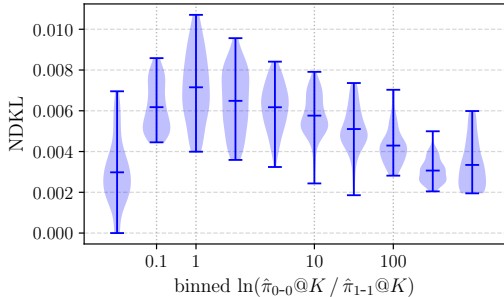

Figure 2: **Left:** permutation-invariant $\Delta_{\text{DP}}$ (Eq. 2). **Right:** rank-aware NDKL (Eq. 3). Both metrics are plotted versus binned log-ratio $\ln(\hat{\pi}_{0\text{-}0}/\hat{\pi}_{1\text{-}1})$ at $K$, where $\hat{\pi}_{0\text{-}0}$ and $\hat{\pi}_{1\text{-}1}$ denote the exposure mass assigned to $\mathbb{E}_{0\text{-}0}$ and $\mathbb{E}_{1\text{-}1}$, resp., showing $\Delta_{\text{DP}}$'s invariance and NDKL's sensitivity to shifts within $\mathbb{E}_{\text{intra}} = \mathbb{E}_{0\text{-}0} \cup \mathbb{E}_{1\text{-}1}$.

Table 2: NDKL@1000 across reproduced baselines and MORAL. $\Delta$NDKL = ours − original. Best in **bold**. Consistent with Mattos et al. (2025), we round to two decimals. OOM is an Out-Of-Memory error.

| Model | Metric | CREDIT | FACEBOOK | GERMAN | NBA | POKEC-N | POKEC-Z |
|---|---|---|---|---|---|---|---|
| FairAdj | NDKL$_{\text{ours}}$ ↓ | OOM | $0.02 \pm 0.00$ | $\mathbf{0.01 \pm 0.00}$ | $0.04 \pm 0.02$ | OOM | OOM |
| | $\Delta$NDKL ↓ | OOM | $(-0.08)$ | $(-0.09)$ | $(-0.07)$ | OOM | OOM |
| FairWalk | NDKL$_{\text{ours}}$ ↓ | $0.02 \pm 0.01$ | $0.02 \pm 0.01$ | $0.03 \pm 0.01$ | $0.03 \pm 0.01$ | $0.03 \pm 0.01$ | $0.03 \pm 0.01$ |
| | $\Delta$NDKL ↓ | $(-0.04)$ | $(-0.04)$ | $(-0.08)$ | $(-0.03)$ | $(-0.04)$ | $(-0.04)$ |
| DetConstSort | NDKL$_{\text{ours}}$ ↓ | $0.02 \pm 0.01$ | $0.02 \pm 0.00$ | $0.02 \pm 0.00$ | $\mathbf{0.01 \pm 0.00}$ | $0.02 \pm 0.00$ | $0.03 \pm 0.01$ |
| | $\Delta$NDKL ↓ | $(-0.04)$ | $(-0.13)$ | $(-0.02)$ | $(-0.08)$ | $(-0.05)$ | $(-0.20)$ |
| **MORAL** | NDKL$_{\text{ours}}$ ↓ | $\mathbf{0.00 \pm 0.00}$ | $\mathbf{0.01 \pm 0.00}$ | $\mathbf{0.01 \pm 0.00}$ | $\mathbf{0.01 \pm 0.00}$ | $\mathbf{0.01 \pm 0.00}$ | $\mathbf{0.01 \pm 0.00}$ |
| | $\Delta$NDKL ↓ | $(-0.04)$ | $(0.00)$ | $(-0.02)$ | $(-0.01)$ | $(-0.02)$ | $(-0.03)$ |

Table 3: Precision@1000 across reproduced baselines and MORAL. $\Delta$Precision = ours − original. Best in **bold**. Consistent with Mattos et al. (2025), we round to two decimals. OOM is an Out-Of-Memory error.

| Model | Metric | CREDIT | FACEBOOK | GERMAN | NBA | POKEC-N | POKEC-Z |
|---|---|---|---|---|---|---|---|
| FairAdj | Precision$_{\text{ours}}$ ↑ | OOM | $0.94 \pm 0.01$ | $\mathbf{0.99 \pm 0.00}$ | $0.52 \pm 0.03$ | OOM | OOM |
| | $\Delta$Precision ↑ | OOM | $(+0.52)$ | $(+0.45)$ | $(+0.02)$ | OOM | OOM |
| FairWalk | Precision$_{\text{ours}}$ ↑ | $\mathbf{1.00 \pm 0.00}$ | $0.92 \pm 0.01$ | $0.85 \pm 0.01$ | $0.53 \pm 0.01$ | $\mathbf{1.00 \pm 0.00}$ | $\mathbf{1.00 \pm 0.00}$ |
| | $\Delta$Precision ↑ | $(0.00)$ | $(-0.04)$ | $(-0.09)$ | $(-0.02)$ | $(0.00)$ | $(0.00)$ |
| DetConstSort | Precision$_{\text{ours}}$ ↑ | $\mathbf{1.00 \pm 0.00}$ | $\mathbf{0.97 \pm 0.00}$ | $0.95 \pm 0.00$ | $\mathbf{0.75 \pm 0.01}$ | $\mathbf{1.00 \pm 0.00}$ | $\mathbf{1.00 \pm 0.00}$ |
| | $\Delta$Precision ↑ | $(+1.00)$ | $(+0.97)$ | $(+0.40)$ | $(+0.54)$ | $(+0.93)$ | $(+0.99)$ |
| **MORAL** | Precision$_{\text{ours}}$ ↑ | $\mathbf{1.00 \pm 0.00}$ | $\mathbf{0.97 \pm 0.00}$ | $0.95 \pm 0.01$ | $\mathbf{0.75 \pm 0.01}$ | $\mathbf{1.00 \pm 0.00}$ | $\mathbf{1.00 \pm 0.00}$ |
| | $\Delta$Precision ↑ | $(0.00)$ | $(+0.02)$ | $(-0.01)$ | $(-0.05)$ | $(+0.02)$ | $(+0.02)$ |

**Claim C3.** Claim **C3** states that MORAL consistently mitigates previously undetected exposure biases, as measured by NDKL, while keeping its competitive utility relative to baselines, as measured by Precision.

Tables 2 and 3 report NDKL and Precision at $K = 1000$ on six datasets for MORAL and the baselines (all introduced in Section 3.5). For each method, we compare the reproduced scores with the originally reported results of Mattos et al. (2025) (listed in completion in Appendix H). Relative to the original scores, the NDKL is reproduced or improved, whereas Precision improves in most cases with some exceptions, e.g., for FairWalk. The small standard deviations tell us these fairness-utility patterns are consistent across runs. We also observe anomalies in the original results, e.g., below-chance Precision scores for DetConstSort, and that FairAdj runs out of memory on three datasets, whereas MORAL remains tractable memory-wise.

Relative to the reproduced scores of the in-processing baselines (FairAdj, FairWalk) and the post-processing baseline (DetConstSort), the bold entries show that MORAL consistently achieves (tied-)lowest, near-zero NDKL. At the same time, the reduction in NDKL/exposure bias comes with a minimal trade-off in utility:

MORAL achieves the (near-)highest Precision on all datasets. The trade-off in utility is most visible on the NBA dataset, whose smaller scale (both in number of nodes $|\mathbb{V}|$ and number of edges $|\mathbb{E}|$, see Appendix C.1) makes top-$K$ precision more sensitive to exposure-preserving re-rankings because cut-off $K$ spans a larger fraction of the candidate pools. MORAL and DetConstSort are indistinguishable in Precision to two decimals, consistent with both methods post-processing/re-ranking the same base scores and largely preserving the top-$K$ true positives while re-distributing exposure. Overall, MORAL reduces exposure disparities while largely preserving top-$K$ true positives, supporting Claim **C3**.

## 4.2 Results beyond original paper

We conducted three additional analyses to further evaluate the paper's claims: an asymmetric homophily stress-test, a metric robustness analysis, and testing categorical sensitive attributes.

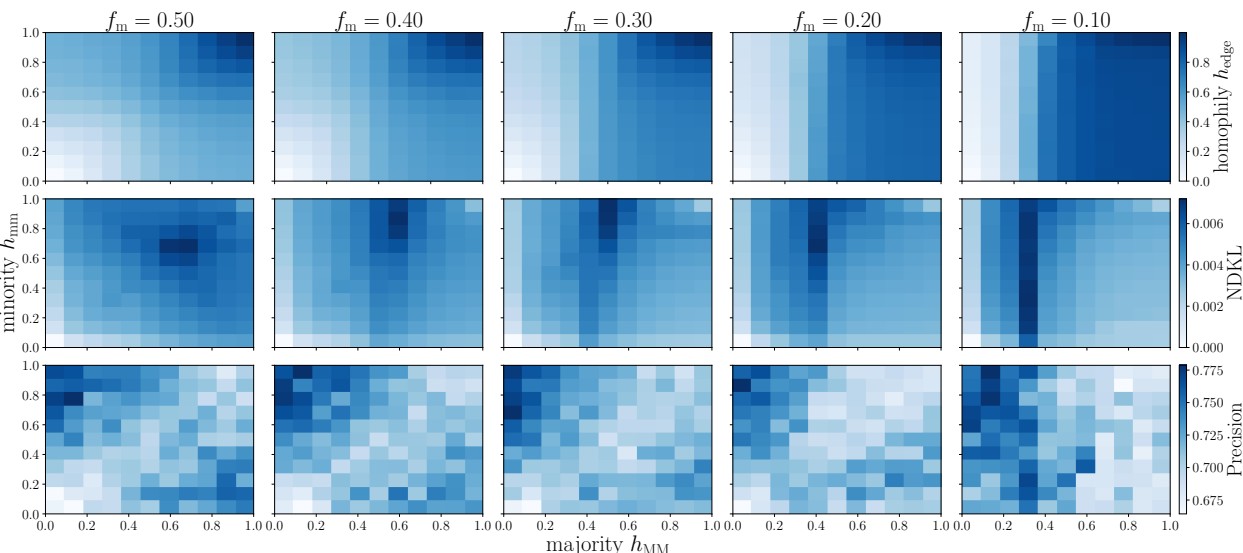

Figure 3: Homophily $h_{\text{edge}}$, NDKL, and Precision on heatmaps of $h_{\text{MM}}$ and $h_{\text{mm}}$ over five $f_{\text{m}}$ values.

**Asymmetric homophily stress-test.** Figure 3 shows a heatmap for $h_{\text{edge}}$, NDKL, and Precision as function of $h_{\text{mm}}$, $h_{\text{MM}}$, and $f_{\text{m}}$. We first observe that $h_{\text{edge}}$ increases with $h_{\text{mm}}$ and $h_{\text{MM}}$; as $f_{\text{m}}$ decreases, the increase becomes more dominated by $h_{\text{MM}}$. The patterns in the NDKL and Precision show some consistency over the range of $f_{\text{m}}$ values too. For instance, NDKL maxima shift with $f_{\text{m}}$, as changes in the availability of subgroup-pair types change the $(h_{\text{MM}}, f_{\text{m}})$-configuration that maximally misaligns exposure.

To further illuminate these patterns, Figure 4 shows $z$-scored NDKL, AWRF, and Precision, mean-averaged over minority homophily $h_{\text{mm}}$ for each $(h_{\text{MM}}, f_{\text{m}})$-configuration; in other words, it projects the Figure 3 heatmaps onto the $h_{\text{MM}}$ axis. Across $f_{\text{m}}$ values, NDKL and AWRF peaks coincide with $h_{\text{edge}} \approx 0.6$, even though the peak's location in $h_{\text{MM}}$ shifts as $f_{\text{m}}$ varies. This medium-homophily setting yields the largest exposure divergence since within-group edges are frequent and rankings are diverse. At higher homophily, the NDKL and AWRF decrease again: subgroup-pair candidate pools become smaller, reducing the degrees of freedom for exposure redistribution and thereby for divergence. For a different reason, Precision tends to drop with increasing $h_{\text{edge}}$: in high homophily, MORAL's decoupled predictors receive less training signal. Finally, Appendix E distinguishes absolute homophily $h_{\text{edge}}$ from relative homophily $\Delta h$, showing that exposure-based fairness metrics respond primarily to absolute homophily. Together, these results show that MORAL's fairness behaviour and performance respond predictably to the graph's homophily structure.

Comparing the synthetic heatmaps with Tables 2 and 3, NDKL appears to be of similar magnitude in both the synthetic and real-data setting. In contrast, Precision lies in a substantially lower range in the synthetic setting. Most plausibly, this can be explained by the difference in features between the scenarios: the DPAH features are non-informative with Gaussian i.i.d. features ($d = 16$), whereas the used datasets have real features of higher dimensionality (e.g., $d = 95$ for NBA), likely making prediction substantially easier.

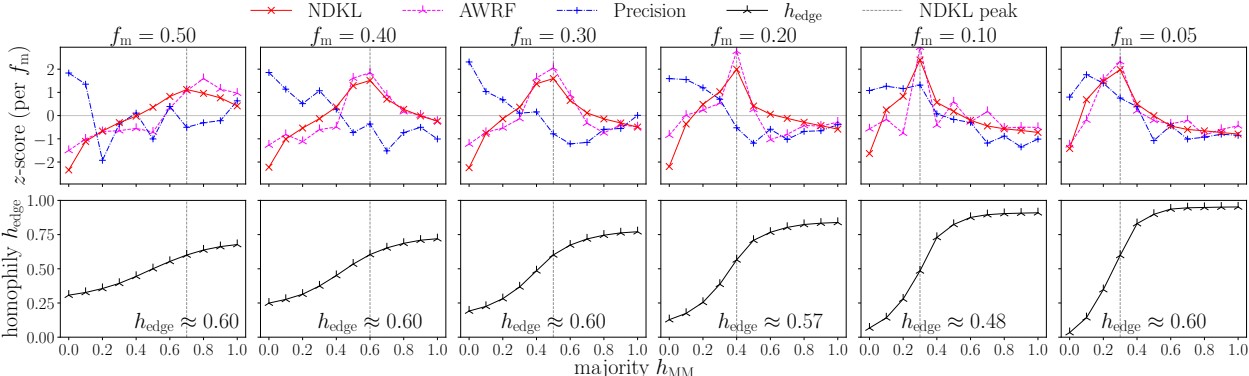

Figure 4: **Top:** $z$-scored NDKL, AWRF, and Precision versus majority homophily $h_{\text{MM}}$ for decreasing minority fraction $f_{\text{m}}$, mean-averaged across minority homophily $h_{\text{mm}}$. **Bottom:** corresponding edge homophily $h_{\text{edge}}$. Vertical dotted lines indicate NDKL peaks, with the corresponding $h_{\text{edge}}$ value annotated.

**Metric robustness analysis.** Table 4 reports the alternative metrics AWRF and NDCG for MORAL and the reproduced baselines. Tables 2 and 3 show the NDKL and Precision scores, respectively. Both NDKL and AWRF remain low across datasets, with the NDKL staying in a 0.00–0.01 range and the AWRF in a 0.001–0.003 range, indicating consistently reduced exposure bias. Figure 4 further shows their coupled reactivity to homophily $h_{\text{edge}}$. Their coupling is expected since both measure ranking-induced exposure relative to $\boldsymbol{\pi}$, though NDKL aggregates position-weighted prefix KL-divergences (Eq. 3) while AWRF applies a single $\ell_1$-norm to the top-$K$ exposure distribution (Eq. 7). Although MORAL directly optimizes the divergence underlying NDKL, the aligned behaviour of AWRF suggests that MORAL is not merely "gaming" the metric. In fact, the separation between MORAL and baselines is larger under AWRF. Regarding utility, the gap between NDCG and Precision on the Facebook, German, and NBA datasets suggests that MORAL preserves ranking quality at higher ranks, which is critical for downstream applications. The gap is largest on NBA since its smaller scale leaves more borderline errors around rank $K$, thus increasing sensitivity to re-rankings. Lastly, Appendix G isolates the effect of MORAL's greedy KL-reranking relative to the raw outputs of the decoupled predictors, showing a 66.9% reduction in NDKL and a 92.6% reduction in AWRF after reranking. Appendix D illustrates fairness (NDKL@$k$, AWRF@$k$) and utility (Precision@$k$, NDCG@$k$) as function of cut-off $k$, showing the found fairness-utility patterns are not specific to $K = 1000$.

Table 4: AWRF↓ and NDCG↑ at $K = 1000$ across reproduced baselines and MORAL (see Tables 2 and 3 for NDKL@1000 and Precision@1000). Best values are underlined. OOM indicates an Out-Of-Memory error.

| Metric | Model | CREDIT | FACEBOOK | GERMAN | NBA | POKEC-N | POKEC-Z |
|---|---|---|---|---|---|---|---|
| **AWRF** | FairAdj | OOM | $0.0421 \pm 0.0139$ | $0.0159 \pm 0.0015$ | $0.0808 \pm 0.0223$ | OOM | OOM |
| | FairWalk | $0.0111 \pm 0.0060$ | $0.0288 \pm 0.0113$ | $0.0518 \pm 0.0223$ | $0.0837 \pm 0.0456$ | $0.0619 \pm 0.0234$ | $0.0353 \pm 0.0148$ |
| | DetConstSort | $0.0371 \pm 0.0083$ | $0.0111 \pm 0.0037$ | $0.0144 \pm 0.0046$ | $0.0222 \pm 0.0052$ | $0.1225 \pm 0.0027$ | $0.1160 \pm 0.0039$ |
| | **MORAL** | $\underline{0.0021 \pm 0.0000}$ | $\underline{0.0032 \pm 0.0000}$ | $\underline{0.0021 \pm 0.0000}$ | $\underline{0.0020 \pm 0.0000}$ | $\underline{0.0023 \pm 0.0000}$ | $\underline{0.0012 \pm 0.0000}$ |
| **NDCG** | FairAdj | OOM | $0.9934 \pm 0.0025$ | $\underline{0.9993 \pm 0.0003}$ | $0.8799 \pm 0.0362$ | OOM | OOM |
| | FairWalk | $0.9999 \pm 0.0001$ | $0.9907 \pm 0.0021$ | $0.9810 \pm 0.0003$ | $0.8752 \pm 0.0135$ | $\underline{0.9999 \pm 0.0001}$ | $\underline{0.9995 \pm 0.0004}$ |
| | DetConstSort | $\underline{1.0000 \pm 0.0000}$ | $0.9951 \pm 0.0004$ | $0.9942 \pm 0.0011$ | $\underline{0.9608 \pm 0.0025}$ | $0.9995 \pm 0.0006$ | $\underline{0.9995 \pm 0.0003}$ |
| | **MORAL** | $\underline{1.0000 \pm 0.0000}$ | $\underline{0.9960 \pm 0.0004}$ | $0.9942 \pm 0.0012$ | $0.9598 \pm 0.0035$ | $0.9995 \pm 0.0006$ | $\underline{0.9995 \pm 0.0003}$ |

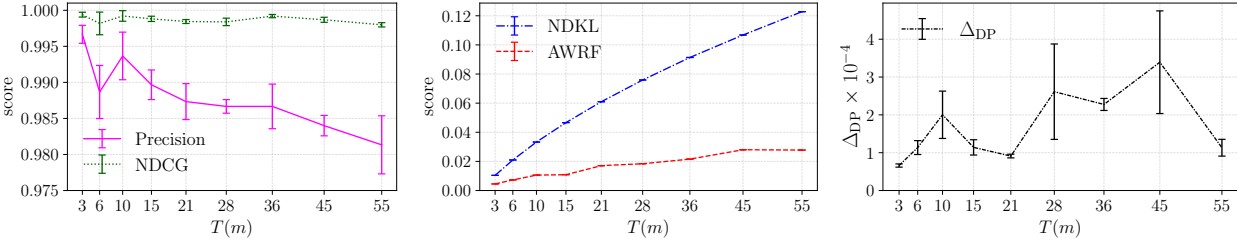

Figure 5: Metrics (mean $\pm$ standard deviation) vs. the number of subgroup-pair types $T(m) = \frac{m(m+1)}{2}$ on the Credit dataset. **Left:** Utility (Precision & NDCG). **Middle:** Fairness (NDKL & AWRF). **Right:** $\Delta_{\text{DP}}$.

**Categorical sensitive attributes.** Figure 5 shows MORAL's performance on the Credit dataset up to cardinality $m = 10$, equivalent to $T(10) = 55$ subgroup-pair types. $\Delta_{\mathrm{DP}}$ remains on the order of $10^{-4}$ with fluctuations but no systematic dependence on $T$, as it compares aggregated exposure mass and is invariant to its distribution within $\mathbb{E}_{\mathrm{intra}}$ and $\mathbb{E}_{\mathrm{inter}}$. In contrast, NDKL and AWRF degrade monotonically, as increasing $T$ enlarges the set of subgroup-pair types over which exposure must be balanced, making simultaneous fairness over all types progressively more difficult. NDKL increases more steeply than AWRF because it aggregates KL-divergences for every $k$th prefix, turning stepwise deviations of $\hat{\boldsymbol{\pi}}_k$ from $\boldsymbol{\pi}$ into an accumulated penalty that grows with $T$. AWRF instead measures a single $\ell_1$-norm at $K$ (i.e., no prefix-by-prefix accumulation) (with the $\ell_1$-norm additionally operating on a different scale than KL-divergence (Cover & Thomas, 2006)).

Utility decreases slightly as $T$ increases but remains high (in the range of 0.97–1.00). The widening gap between Precision and NDCG suggests that top-ranked edges remain accurate while lower-ranked positions become less stable with higher $T$, consistent with the earlier observations on the NBA dataset results. Overall, MORAL maintains utility well at higher cardinality, but the observed NDKL and AWRF degradation indicates that aligning exposure with $\boldsymbol{\pi}$ becomes more challenging as $T$ increases.

## 5 Discussion

The primary aim of this study was to reproduce and evaluate the three main claims by Mattos et al. (2025), which we introduced in Section 2. In Section 4.1, we found our results support Claims **C1**–**C3**. Across all datasets, we reproduced or even improved upon the originally reported fairness-utility trade-offs. We attribute the discrepancies between the original results and ours primarily to underspecification in the released codebase; in cases of ambiguity, we prioritized optimization, leading to improved fairness and utility. Compared to in- and post-processing baselines, MORAL performed best, suggesting that enforcing distribution-preserving exposure at ranking time is more effective than training-time interventions or heuristic post-processing. Additionally, we confirm generalizability to additional datasets in Appendix F. Beyond reproducing the main results, our extensions assessed the generality and limits of the framework.

Firstly, the asymmetric homophily stress-test clarified when MORAL's trade-offs are most evident. In Figures 3 and 4, we observe that exposure-based metrics are most strained at intermediate *absolute* homophily (rather than *relative* homophily, see Appendix E): high cross-group connectivity forces allocation of limited exposure over many competing subgroup-pair pools, making exposure disparities easier to surface. At higher homophily, subgroup-pair candidate pools become structurally constrained and the learning signal for some decoupled predictors weakens, consistent with declining utility. Taken together, MORAL performs strongly and consistently across the tested homophily settings, with homophily inducing only mild variation around already low exposure disparity and high utility, supporting Claim **C3**. Notably, the peaks of the exposure-based metrics were established under controlled synthetic conditions, where features are i.i.d. and mixing is isolated from feature design; in real graphs, features, degree effects, and group structure are typically entangled, so the exact location may shift. This concerns the transferability of the peak's location from synthetic to real graphs, not the relevance of absolute homophily as a signal: since NDKL and AWRF respond consistently to absolute homophily regardless of the specific mixing configuration, we view this as a structural effect of homophily rather than a generator artefact. Appendix E further shows that $\Delta_{\mathrm{DP}}$ is largely insensitive to homophily, whereas NDKL varies substantially, thus reinforcing Claims **C1** & **C2**.

Secondly, we assessed whether our conclusions are metric-dependent by complementing NDKL and Precision with AWRF and NDCG. The larger separation between MORAL and the baselines under AWRF suggests MORAL's reductions in exposure bias are not NDKL-specific nor a by-product of optimizing a closely-aligned objective, but that it reduces exposure disparity more generally. We note, however, that this robustness is demonstrated within exposure-based fairness metrics and does not extend to fundamentally different fairness definitions. The NDCG results, together with the cut-off analysis in Appendix D, suggest that MORAL preserves ranking quality across cut-offs. This includes ranking quality near the top of the ranking, which is important for downstream applications such as social networks, where small shifts at the top of the ranking can result in large attention and visibility differences (Singh & Joachims, 2018; Ferrara et al., 2022). Overall, the results suggest limited metric sensitivity, thus strengthening Claim **C2** & **C3**.

Thirdly, the categorical-attribute extension assessed MORAL in higher-cardinality settings. For fairness, $\Delta_{\text{DP}}$ remains largely insensitive to growing $T$, whereas the exposure-based metrics increase with $T$, supporting Claim **C1** & **C2**. This also suggests finer-grained structure of subgroups makes distribution-preserving exposure alignment progressively harder, as exposure must be balanced across a growing number of subgroup-pair types. The two exposure-based metrics differ in how they reflect this effect. NDKL rises more sharply, partly because it accumulates prefix-wise deviations, meaning some of its increases are attributable to purely its mechanism, not "fairness" in itself. Subgroup-pair-adapted AWRF provides a complementary single-shot, position-weighted perspective on exposure divergence and increases less steeply. Utility decreases modestly as $T$ grows. The gap between Precision and NDCG widening with $T$ indicates that errors concentrate lower within the top-$K$ as $T$ grows, while top-of-ranking utility remains high. The decrease in utility is partly due to weaker training signal, similar to what we observed in high homophily. Overall, MORAL behaves meaningfully beyond the binary setting (Claim **C3**), but the results reveal a practical fairness-scalability trade-off: as $T$ grows (and subgroup-pair pools become smaller), meeting a distribution-preserving target $\boldsymbol{\pi}$ becomes harder and decoupled training becomes more computationally costly.

In conclusion, dyadic demographic parity is insufficient for ranked link prediction because it can obscure subgroup-pair exposure disparities revealed by NDKL, suggesting that prior work relying on dyadic parity may under-report such effects. More broadly, fairness criteria should match downstream use: when outputs are consumed as rankings, exposure should be the relevant currency of fairness. Despite inconsistencies in the original implementation, MORAL behaves as intended and remains robust across our stress-tests and extensions, supporting its use as a simple post-processing method for debiasing ranked link predictions.

**Practical recommendations.** We make a few practical recommendations for practitioners: (1) in ranked link prediction, demographic parity is a poor proxy for fairness based on exposure: we recommend to prioritize exposure-based metrics; (2) both NDKL and AWRF are suitable at evaluating for exposure bias, and reporting both is especially useful at higher sensitive-attribute cardinality; (3) use MORAL when outputs are consumed as rankings and top-$K$ exposure is a fairness concern, especially when per-type predictors are feasible; (4) utility trade-offs are largest on smaller graphs, so practitioners should be cautious there and evaluate multiple cut-offs, as lower-ranked positions appear more sensitive to reranking; (5) the cut-off analysis suggests that MORAL's fairness-utility patterns are broadly stable across cut-offs; (6) under the synthetic conditions tested, auditing for exposure bias is most informative at intermediate absolute homophily, making it a reasonable starting point for audits, although the exact location may shift in real graphs; (7) beyond binary attributes, MORAL scales to a more realistic, higher-cardinality setting; however, the granularity should be treated as a fairness-scalability trade-off: exposure alignment becomes harder and more computationally costly at higher cardinality (while top-$K$ utility degrades only modestly).

**Limitations and future work.** A natural direction for future work is to study how compute scales versus fairness and utility. In particular, it remains unclear whether training separate predictors per subgroup-pair type is necessary to achieve strong (or optimal) fairness-utility trade-offs, or whether parameter sharing (e.g., by hierarchical grouping or a shared backbone) can reduce combinatorial growth without sacrificing performance. In Appendix G, we make a start at this by ablating the effect of greedy KL-aggregation. Second, our adapted AWRF uses an $\ell_1$-distance; testing KL-divergence-based variants could further isolate the usability of both NDKL and AWRF.

Finally, as mentioned in Section 3.3, preserving $\boldsymbol{\pi}$ is a design choice rather than a universally correct fairness target (Mitchell et al., 2021). Thus, the low NDKL and AWRF values achieved by MORAL should be interpreted as evidence of strong performance under distribution-preserving fairness, rather than for support of a more general claim about fairness, as the reference distribution $\boldsymbol{\pi}$ might contain biases in itself. Future work should compare fairness objectives that reflect different normative commitments.

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

# A MORAL and NDKL

This Appendix provides the MORAL pseudocode and a brief explanation on our NDKL implementation.

## A.1 MORAL pseudocode

Algorithm 1 outlines the MORAL algorithm pseudocode, adapted from Mattos et al. (2025) but adjusted to our notation. In Section 3.6.2, we touch on the algorithm's similarities to the NDKL metric (Eq. 3).

---

**Algorithm 1** MORAL: Multi-Output Ranking Aggregation for Link Fairness

---

**Input:**

- Candidate ordered sets $\mathbf{C}_j = ((u, v, \text{score}), (u_2, v_2, \text{score}_2), \dots)$ for each group $j \in \{0, 1, 2\}$ (sorted by descending score);

- Target distribution $\boldsymbol{\pi} = (\pi_0, \pi_1, \pi_2)$;

- Total output size $n$.

**Output:** Ranking list $\mathbf{R}$ of $n$ predicted edges with assigned group labels

Initialize exposure counts: $\boldsymbol{c} \leftarrow (0, 0, 0)$  Initialize output ranking: $\mathbf{R} \leftarrow [\,]$

**for** $t \leftarrow 1$ **to** $n$ **do**

    Initialize best objective: min_kl $\leftarrow \infty$, selected_group $\leftarrow -1$, selected_edge $\leftarrow$ None

    **foreach** group $j \in \{0, 1, 2\}$ such that $\mathbf{C}_j$ is not empty **do**

        Let $(u, v, \text{score}) \leftarrow$ top element in $\mathbf{C}_j$

        Temporarily update counts: $c'_j \leftarrow c_j + 1$, $q'_j \leftarrow \frac{c'_j}{t}$, $q'_{j' \neq j} \leftarrow \frac{c_{j'}}{t}$

        Compute KL-divergence: $D_{\text{KL}}(\boldsymbol{q}' \,\|\, \boldsymbol{\pi})$, where $\boldsymbol{q}' = (q'_0, q'_1, q'_2)$

        **if** $D_{\text{KL}}(\boldsymbol{q}' \,\|\, \boldsymbol{\pi}) <$ min_kl **then**

            Update min_kl $\leftarrow D_{\text{KL}}(\boldsymbol{q}' \,\|\, \boldsymbol{\pi})$, selected_group $\leftarrow j$, selected_edge $\leftarrow (u, v)$

        **end**

    **end**

    Append (selected_edge, selected_group) to $\mathbf{R}$

    Remove top element from $\mathbf{C}_{\text{selected\_group}}$

    Update $c_{\text{selected\_group}} \leftarrow c_{\text{selected\_group}} + 1$

**end**

**return** $\mathbf{R}$

---

## A.2 NDKL implementation

Another part that required further specification and optimization for reproduction is the NDKL. After correspondence with the authors, they were kind to send their NDKL implementation. We found their implementation could be optimized further and adapted to better reflect their mathematical definition:

- Firstly, KL-divergence is not symmetric w.r.t. the distributions it evaluates: rearranging the order of distributions — for example from $D_{\text{KL}}(\boldsymbol{\pi}, \hat{\boldsymbol{\pi}}_k)$ to $D_{\text{KL}}(\hat{\boldsymbol{\pi}}_k, \boldsymbol{\pi})$, where $\hat{\boldsymbol{\pi}}_k$ is the empirical edge group distribution up to rank $k$ and $\boldsymbol{\pi}$ the global target distribution — changes the resulting NDKL value. We chose to follow the definition defined in the paper text: $D_{\text{KL}}(\hat{\boldsymbol{\pi}}_k, \boldsymbol{\pi})$.

- Secondly, we removed an extra normalisation term used in their NDKL function, which was not mentioned in the paper text.

- Thirdly, for numerical stability, we smooth the target distribution $\boldsymbol{\pi}$ with clamping and renormalisation, i.e. $\boldsymbol{\pi} = \text{clamp}(\boldsymbol{\pi}, \eta)$, with $\eta = 1 \times 10^{-12}$, followed by $\boldsymbol{\pi} \leftarrow \boldsymbol{\pi} / \sum_j \pi_j$, where $j$ indexes subgroup-pair types. This ensures $\pi_j \geq \eta$ for all $j$ and prevents $D_{\text{KL}}(\hat{\boldsymbol{\pi}}_k, \boldsymbol{\pi})$ from diverging when $\pi_j \approx 0$ with $\hat{\pi}_{k,j} > 0$.

Our implementation can be found in the following file: `https://github.com/Floris93100/reproducing-MORAL/blob/main/scripts/helpers/metrics.py`.

# B    Hyperparameters

This section first details the hyperparameters used for reproduction, followed by the hyperparameters used for the homophily experiments.

## B.1    Hyperparameters for reproduction

The datasets were trained on a graph neural network (Wu et al., 2020) with a GCN (Kipf & Welling, 2016) encoder and a dot-product decoder. Table 5 lists the hyperparameters used for training the model.

Table 5: Model architecture and hyperparameter specifications for the MORAL framework.

| General setup | | Encoder & Decoder | | Training details | |
|---|---|---|---|---|---|
| Hyperparameter | Value | Component | Value | Parameter | Value |
| Datasets | See Table 1 | Encoder | GCN | Optimizer | Adam |
| Seeds | 0, 1, 2 | Hidden dim. | 128 | Learning rate | $3 \times 10^{-4}$ |
| Subgroup-pair types | 3 | Encoding layers | 2 | Weight decay | 0.0 |
| $K$ | 1000 | Decoder | Dot product | Batch size | 1024 |
| Feature normalisation | Min-max | Activation | ReLU | Max epochs | 500 |
| | | Dropout | 0.2 | | |

## B.2    Hyperparameters for homophily experiments

To generate the graphs with differing homophilies, the *Directed Preferential Attachment with Homophily* (DPAH) graph generator by Espín-Noboa et al. (2022) was employed, from whom the hyperparameters in Table 6 were adopted. See Section 3.6.1 for a detailed explanation on the utilisation.

Table 6: Hyperparameter specifications for the DPAH synthetic graph generation.

| Global parameters | | Demographic config. | | Homophily | |
|---|---|---|---|---|---|
| Hyperparameter | Value | Parameter | Value | Parameter | Value |
| Nodes $N$ | 1000 | Minority fraction $f_{\mathrm{m}}$ | [0.05, 0.10, 0.20, 0.30, 0.40, 0.50] | Majority homophily $h_{\mathrm{MM}}$ | [0.0, 1.0] in steps of 0.1 |
| Edge density $d$ | 0.006 | Majority label | 0 (M) | Minority homophily $h_{\mathrm{mm}}$ | [0.0, 1.0] in steps of 0.1 |
| Feature dim. | 16 | Minority label | 1 (m) | PL-Outdegree $\gamma_{\mathrm{M}}$ | 2.5 |
| Generation seed | 42 | Groups ($G$) | {M, m} | PL-Outdegree $\gamma_{\mathrm{m}}$ | 2.5 |
| Graph type | Directed | | | Preferential attachment | In-degree |

## B.3    Baseline experiments

To compare the results, a few baseline models were implemented. See Section 4.1 for the baseline results.

**FairAdj.** FairAdj is an algorithm proposed by Li et al. (2021), the same paper to which the contributions of Mattos et al. (2025) are a response. It empirically learns a fair adjacency matrix by updating the normalized adjacency matrix while maintaining the graph structure. Li et al. (2021)'s implementation uses a Variational Graph Autoencoder (VGAE) with a two-layer Graph Convolutional Network (GCN) as the inference model. The training procedure consists of two alternating phases: $T_1$ epochs for optimizing link prediction and $T_2$ epochs for dyadic fairness. They use projected gradient descent to regulate edge weights. We evaluate with the utility and fairness metrics on the top-$K$ ranking list without applying a post-processing method. We adapted the FairAdj codebase for our implementation and refer to our GitHub repository for further details and references to the original model. We adopted their hyperparameters as seen in Table 7.

Table 7: Hyperparameters and architectural configurations for the FairAdj framework (Li et al., 2021)

| Model architecture | | Optimization (Phase 1: $T_1$) | | Fairness (Phase 1: $T_2$) | |
|---|---|---|---|---|---|
| Hyperparameter | Value | Parameter | Value | Parameter | Value |
| Base model | GCN-VAE | Optimizer | AdaM | Fairness metric | $\Delta_{\mathrm{DP}}$ |
| 1st hidden layer | 32 | Learning rate ($\eta_\theta$) | 0.01 | Learning rate ($\eta_{\tilde{A}}$) | Dataset-specific |
| 2nd hidden layer | 16 | Outer epochs | 4 | $T_2$ sub-epochs | Dataset-specific |
| Activation func. | ReLU | $T_1$ sub-epochs | 50 | PGD projection | Simplex/right stochastic |
| Decoder | Dot product | Weight clipping | 1.0 | Structural constraint | Fixed sparsity |
| Dropout | 0.0 | KL-divergence | Gaussian prior | Gradient descent | Projected |

**FairWalk.** FairWalk is a fairness-aware graph embedding algorithm proposed by Rahman et al. (2019). It is an in-processing method, because it incorporates fairness directly into the embedding learning procedure by adjusting the random walk generation. FairWalk is an extension of node2vec (Mikolov et al., 2013). First, neighbours are grouped by sensitive attribute at each step, after which a group is sampled uniformly and a random neighbour is chosen from that group. Then, the standard skip-gram (word2vec) objective learns node embeddings based on the resulting walks. We evaluate on the outputted top-$K$ edge ranking list. Our implementation is inspired by the authors' official GitHub repository and we adopt the node2vec-style hyperparameters reported in their paper, as shown in Table 8. We refer to our codebase for more details.

Table 8: Hyperparameters and configuration for FairWalk.

| Walk generation | | Skip-gram training | | Experimental setup | |
|---|---|---|---|---|---|
| Parameter | Value | Parameter | Value | Parameter | Value |
| Embedding dimension ($d$) | 128 | Skip-gram (sg) | 1 | Runs | 3 |
| Walk length ($L$) | 80 | Hierarchical softmax (hs) | 0 | Random seed | $\{0, 1, 2\}$ |
| Number of walks per node ($r$) | 20 | Negative samples ($k$) | 5 | | |
| Context window size ($w$) | 10 | Weight decay | 0.0 | | |
| | | Learning rate ($\eta$) | $3 \times 10^{-4}$ | | |
| | | Number of training epochs | 5 | | |

**DetConstSort.** DetConstSort is a deterministic constrained re-ranking method proposed by Geyik et al. (2019). It acts as a post-processor and re-ranks the initial list by prioritizing high-scoring items and enforcing group proportion constraints at each rank position. Due to Geyik et al. (2019) not providing an official public implementation, we base our implementation on the repository by Ghosh et al. (2021) and adapt it to the link prediction setting by treating candidate edges as ranked items and edge-types as groups. We apply DetConstSort on the exact same output ranking produced by the three edge-type specific models that were used in the MORAL training pipeline. Thus, for DetConstSort we do not change the training procedure.

## C   Statistics

This Section provides statistics on the datasets and on the environmental impact of this study.

### C.1   Dataset statistics

Table 9 gives more dataset statistics in supplement of Table 1 in Section 3.4, with gradients per epoch as

$$\text{gradients per epoch} = \frac{|\mathbb{E}_{\text{train}}|}{|\mathbb{S}| \cdot \text{batch size} \cdot \binom{|\mathbb{S}|}{2}},$$

where $|\mathbb{S}|$ is the sensitive attribute cardinality and $|\mathbb{E}_{\text{train}}|$ is the number of training edges (Mattos et al., 2025). We replicate Mattos et al. (2025)'s statistics (in their *Table 1*) except for slight deviations in $|\mathbb{E}|$ (of maximum 3 edges), probably due to different preprocessing, e.g., (missing) deduplication. Table 10 shows homophily statistics for the datasets and classifies the datasets into homophily levels.

### C.2 Environmental impact statistics

The total computational expense for reproducing the original experiments equals 17.96 GPU hours with an estimated 1.24 $kgCO_2eq$. For reference, the estimated carbon intensity of the Netherlands was 411 $g\,CO_2eq\,kWh^{-1}$ in January 2026 (Nowtricity, 2026). Table 11 breaks these numbers down further. The total computational expense for the homophily extension equals 30.41 GPU hours with an estimated 1.41 $kgCO_2eq$. Emissions were estimated using CodeCarbon (Lacoste et al., 2023).

By summing these estimated emissions, we get a total of 2.65 $kgCO_2eq$ for the above experiments, which, by taking the recent estimated carbon intensity per kWh of the Netherlands into our calculation, is equivalent to heating $\pm 70$ liters of water from $20\,°C$ to $100\,°C$.

Table 9: Dataset statistics including edge-type proportions. The upper six datasets are the datasets used by Mattos et al. (2025) and the lower two are added for testing different topologies, see Appendix F. GARN stands for Global Airport Routes Network. GpE denotes the gradients per epoch.

| Dataset | $|\mathbb{V}|$ | $|\mathbb{E}|$ | Feature dim. | Attribute | $\mathbb{E}_{0\text{-}1}$ | $\mathbb{E}_{1\text{-}1}$ | $\mathbb{E}_{0\text{-}0}$ | Topology | GpE |
|---|---|---|---|---|---|---|---|---|---|
| CREDIT | 30000 | 96162 | 13 | Age | 12% | 87% | 1% | Periphery | 47 |
| FACEBOOK | 1045 | 18723 | 573 | Gender | 42% | 44% | 13% | Periphery | 10 |
| GERMAN | 1000 | 15218 | 27 | Gender | 20% | 61% | 19% | Periphery | 8 |
| NBA | 403 | 7434 | 95 | Nationality | 28% | 63% | 9% | Periphery | 4 |
| POKEC-N | 66569 | 361932 | 276 | Gender | 54% | 23% | 22% | Community | 177 |
| POKEC-Z | 67796 | 432569 | 265 | Gender | 5% | 58% | 37% | Community | 212 |
| GARN | 7698 | 18858 | 5 | Hub/non-hub | 27% | 68% | 5% | Core-periphery | 7 |
| CHAMELEON | 890 | 8854 | 2326 | See Section F | 50% | 14% | 35% | Flat | 4 |

Table 10: Edge homophily relative to random mixing across datasets. We report the observed edge homophily $h_{\text{edge}}$ (Eq. 4), the expected homophily under random mixing $h_{\text{edge}}^{\text{rand}}$ (Eq. 9), and the excess homophily $\Delta h = h_{\text{edge}} - h_{\text{edge}}^{\text{rand}}$ (Eq. 10). We classify $\Delta h$ as homophilic ($\Delta h > 0$) or heterophilic ($\Delta h < 0$) relative to random mixing, with low/medium/high categories based on $|\Delta h|$.

| Dataset | $h_{\text{edge}}^{\text{rand}}$ | $h_{\text{edge}}$ | $\Delta h$ | Homophily type |
|---|---|---|---|---|
| CREDIT | 0.8370 | 0.8790 | 0.0420 | Low homophily |
| FACEBOOK | 0.5502 | 0.5757 | 0.0255 | Low homophily |
| GERMAN | 0.5722 | 0.8048 | 0.2326 | High homophily |
| NBA | 0.6100 | 0.7237 | 0.1137 | Medium homophily |
| POKEC-N | 0.5003 | 0.4560 | -0.0443 | Low heterophily |
| POKEC-Z | 0.5001 | 0.4507 | -0.0494 | Low heterophily |
| GLOBAL AIRPORT ROUTES NETWORK | 0.8162 | 0.7301 | -0.0861 | Medium heterophily |
| CHAMELEON | 0.5730 | 0.4955 | -0.0775 | Medium heterophily |

Table 11: Environmental impact metrics for reproducing the core link prediction models, each trained on 500 epochs, on six different datasets. Mean and standard deviation over 3 seeds are reported.

| Dataset | $kgCO_2eq$ | Total $kgCO_2eq$ | kWh | Total kWh | Duration (min) |
|---|---|---|---|---|---|
| CREDIT | $0.0200 \pm 0.0059$ | 0.1198 | $0.0746 \pm 0.0221$ | 0.4477 | $22.39 \pm 0.04$ |
| FACEBOOK | $0.0021 \pm 0.0000$ | 0.0064 | $0.0080 \pm 0.0000$ | 0.0240 | $3.24 \pm 0.00$ |
| GERMAN | $0.0029 \pm 0.0001$ | 0.0087 | $0.0109 \pm 0.0003$ | 0.0326 | $2.67 \pm 0.00$ |
| NBA | $0.0023 \pm 0.0001$ | 0.0070 | $0.0087 \pm 0.0002$ | 0.0261 | $2.22 \pm 0.00$ |
| POKEC-N | $0.1476 \pm 0.0223$ | 0.4428 | $0.5515 \pm 0.0833$ | 1.6544 | $134.66 \pm 0.35$ |
| POKEC-Z | $0.2190 \pm 0.0006$ | 0.6569 | $0.8182 \pm 0.0022$ | 2.4545 | $171.63 \pm 0.63$ |

# D  Sensitivity of MORAL to ranking cut-off $K$

To assess sensitivity to the evaluation cutoff, Figure 6 and 7 show fairness and utility as function of $k$, varied from 10 to 1000. The plots show that MORAL's fairness behaviour is robust to the choice of cut-off. On all datasets, NDKL and AWRF drop rapidly for small $k$ and then stabilize near zero, indicating progressively improved exposure alignment as longer prefixes of the ranking are considered. The curves drop quickly since with lower $k$, individual (wrong) predictions have more proportional influence. At the same time, Precision and NDCG remain largely stable on Credit, Facebook, German, and the Pokec datasets, suggesting that the gains in fairness are not accompanied by substantial utility loss. NBA is the exception and has more variation in both utility curves, which is consistent with the stronger sensitivity of this smaller dataset, as previously discussed in Section 4.2.

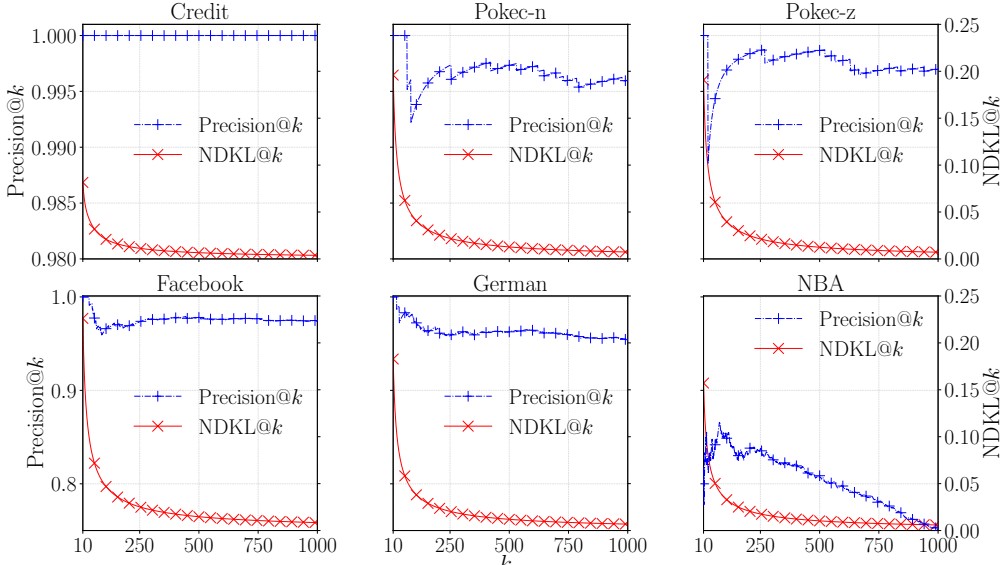

Figure 6:  Precision@$k$ (left axis, blue) and NDKL@$k$ (right axis, red) of the final MORAL ranking as function of cut-off $k$, averaged over three seeds. Axes are aligned horizontally.

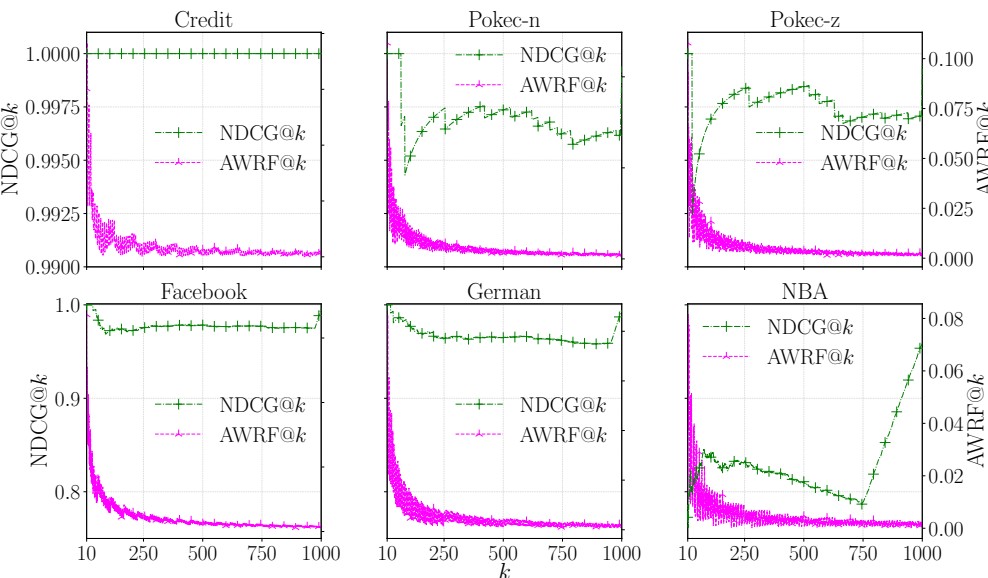

Figure 7:  NDCG@$k$ (left axis, green) and AWRF@$k$ (right axis, magenta) of the final MORAL ranking as function of cut-off $k$, averaged over three seeds. Axes are aligned horizontally.

# E  Additional homophily results

In this section, we support Section 4.2 with additional results. Figure 8 shows a heatmap for $h_{\text{edge}}$, $h_{\text{edge}}^{\text{rand}}$, $\Delta h_{\text{edge}}$, $\Delta_{\text{DP}}$, NDKL, Precision, AWRF, and NDCG, as a function of $h_{\text{mm}}$, $h_{\text{MM}}$, and $f_{\text{m}}$. Here, by Newman (2003)'s random-mixing method (and substituting with $f_{\text{m}}$), we define the expected homophily under random mixing as

$$h_{\text{edge}}^{\text{rand}} = f_{\text{m}}^2 + (1 - f_{\text{m}})^2. \tag{9}$$

Consequently, the relative/excess homophily is

$$\Delta h = h_{\text{edge}} - h_{\text{edge}}^{\text{rand}}. \tag{10}$$

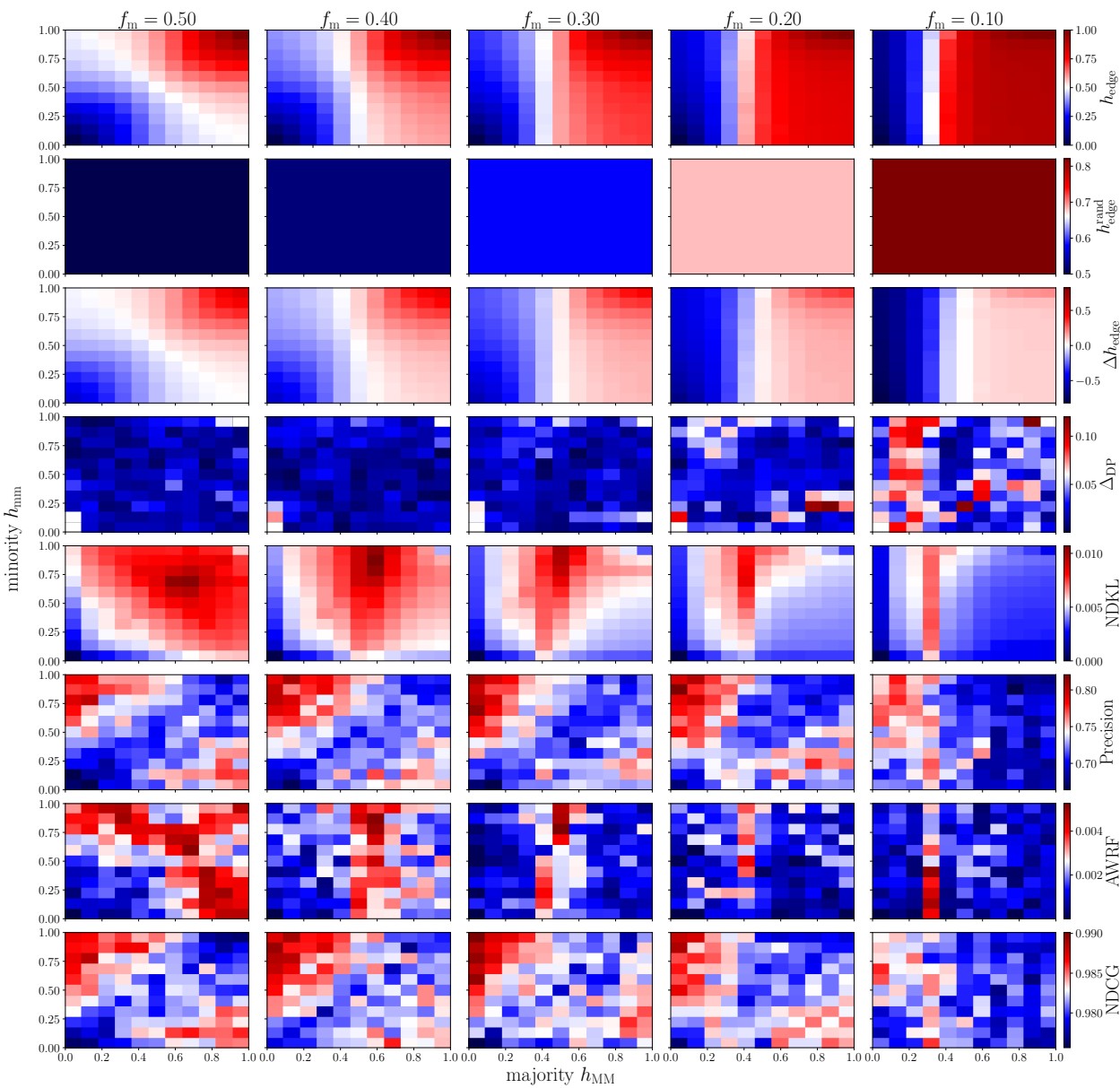

Figure 8: Edge homophily $h_{\text{edge}}$ (Eq. 4), random homophily baseline $h_{\text{edge}}^{\text{rand}}$ dependent on minority fraction $f_{\text{m}}$, homophily w.r.t baseline $\Delta h = h_{\text{edge}} - h_{\text{edge}}^{\text{rand}}$, $\Delta_{\text{DP}}$ (Eq. 2), NDKL (Eq. 3), Precision, AWRF (Eq. 7), and NDCG as a function of minority parameter $h_{\text{mm}}$ and majority parameter $h_{\text{MM}}$, for different values of $f_{\text{m}}$, all at $K = 1000$. As seen by the top row, absolute homophily generally increases with $f_{\text{m}}$ and $h_{\text{MM}}$.

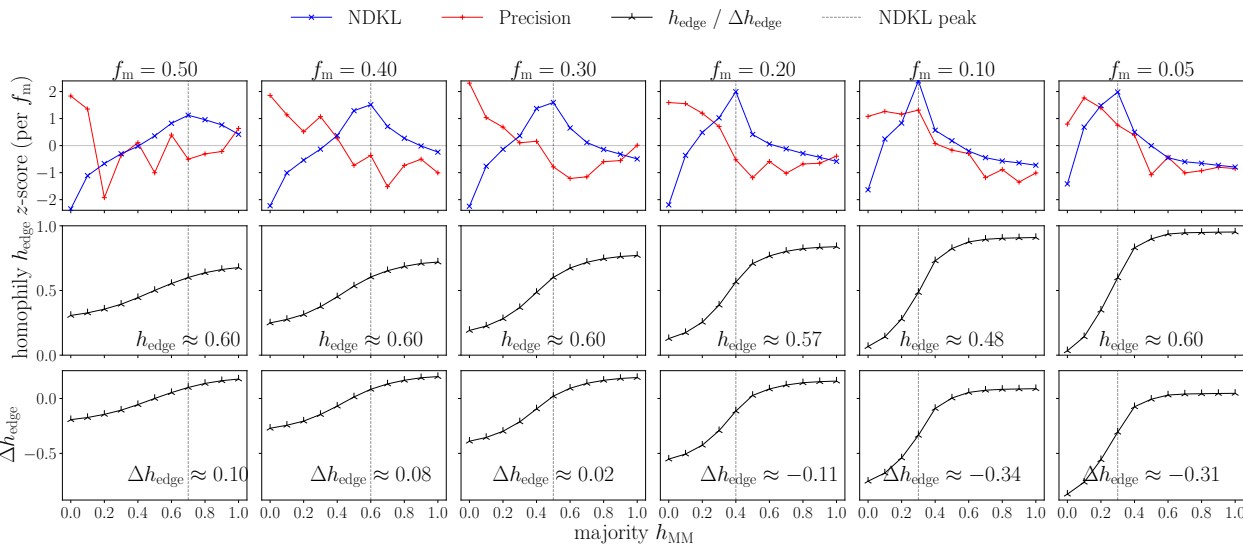

Figure 9: **Top:** $z$-scored NDKL and Precision versus majority homophily $h_{\text{MM}}$ for decreasing minority fraction $f_{\text{m}}$, mean-averaged across minority homophily $h_{\text{mm}}$. **Middle:** corresponding edge homophily $h_{\text{edge}}$. Vertical dotted lines indicate NDKL peaks, with the corresponding $h_{\text{edge}}$ value annotated. **Bottom:** corresponding relative edge homophily $\Delta h_{\text{edge}}$.

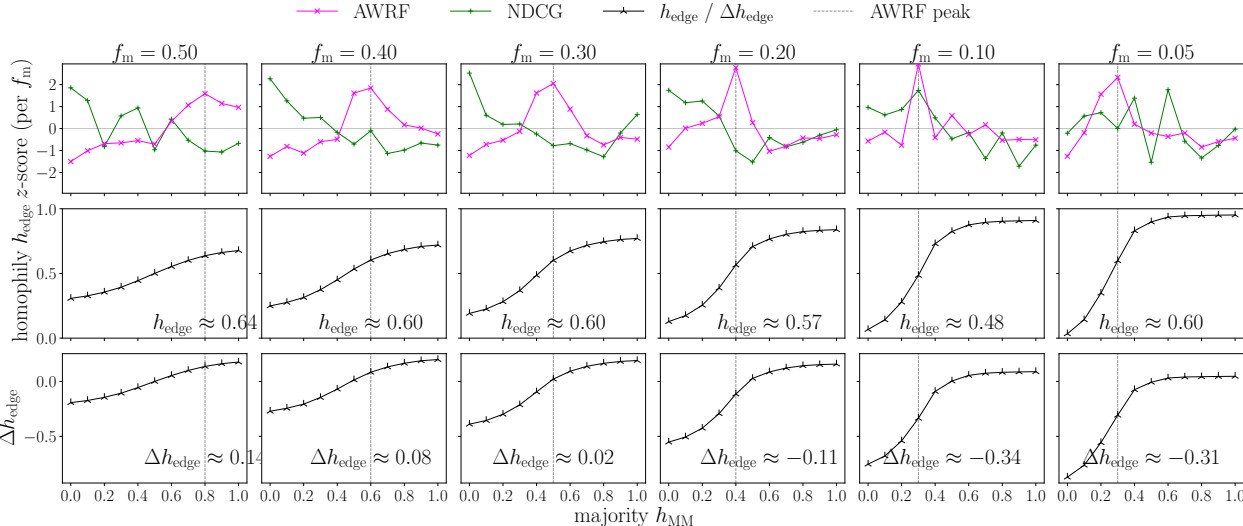

Figure 10: **Top:** $z$-scored AWRF and NDCG versus majority homophily $h_{\text{MM}}$ for decreasing minority fraction $f_{\text{m}}$, mean-averaged across minority homophily $h_{\text{mm}}$. **Middle:** corresponding edge homophily $h_{\text{edge}}$. Vertical dotted lines indicate AWRF peaks, with the corresponding $h_{\text{edge}}$ value annotated. **Bottom:** corresponding relative edge homophily $\Delta h_{\text{edge}}$.

In the heatmaps, we first observe that the absolute edge homophily $h_{\text{edge}}$ increases as $h_{\text{MM}}$ and $h_{\text{mm}}$ increase. As Eq. 9 takes just $f_{\text{m}}$, the random-mixing baseline $h_{\text{edge}}^{\text{rand}}$ heatmap is constant for each minority fraction $f_{\text{m}}$. Therefore, $\Delta h$ highlights where the observed homophily is above or below the random mixing baseline. We see that $\Delta_{\text{DP}}$ reacts comparatively little and inconsistently to increases of $h_{\text{edge}}$.

Figures 9 and 10 show a similar pattern. NDKL and AWRF increase initially (indicating more unfair ranking predictions), but drop once absolute homophily reaches around $h_{\text{edge}} \approx 0.60$ (in most cases). Importantly,

this turning point does not align with a fixed value of the relative homophily $\Delta h$. This implies that these exposure-based fairness metrics are more closely tied to absolute homophily than to relative homophily.

A plausible explanation is that absolute homophily directly shapes the composition of the top of the ranking. When edges concentrate more strongly within groups, a majority-with-majority structure can dominate the highest scores and positions, which is exactly where exposure metrics are most sensitive (Singh & Joachims, 2018) (for example by following the logarithmic decay of Eq. 1). A normalized measure such as $\Delta h$ can miss this, because it corrects for group size and can take similar values under completely different absolute mixing patterns.

Although in the literature relative homophily is used to compare mixing patterns across group sizes (e.g., the assortativity perspective and inbreeding homophily) (McPherson et al., 2001; Newman, 2003; Currarini et al., 2010; Platonov et al., 2023a), our results suggest that exposure-based fairness metrics in ranked link prediction are primarily sensitive to absolute homophily $h_{\text{edge}}$ rather than to relative homophily $\Delta h$. In particular, the aforementioned NDKL and AWRF peaks occur at similar $h_{\text{edge}}$-levels across settings while occuring at different $\Delta h$-levels, thus implying that comparable $\Delta h$ can yield different exposure (disparities) and visibility at the top of the ranking. Thus, absolute homophily gives a more informative signal than relative homophily for explaining exposure-based fairness in rankings.

## F  Verifying generalizability to additional datasets

In this Appendix Section, we verify the generalizability of MORAL to two additional real-world datasets. Whereas the original datasets, see Section 3.4, had *Community* and *Periphery* topologies, the Global Airport Routes Network (GARN) dataset has a *Core-periphery* topology such that we can test generalizability to a different topology. With the second dataset, Chameleon, we complement our synthetic homophily experiments with a realistic alternative and show MORAL's fairness there is not an artefact of the generator.

**Designing a binary sensitive attribute for GARN.**  For GARN (Peixoto, 2020), the goal is to design the sensitive attribute such that several cores will form with peripheral nodes around it. Airport networks naturally resemble a core-periphery topology (Gallagher et al., 2021; Diop et al., 2023). One can argue for the need of fair link prediction in airport networks by economic reasons, as, e.g. through "rich-get-richer" effects (Subramonian et al., 2023), smaller airports could be disadvantaged in route planning recommendations.

By sorting the airports in descending order by degree, we get a power-law decaying pattern (Guimera & Amaral, 2004). As Guimera & Amaral (2004) point out, the most connected airports are typically not the most central ones. Hence, proper classification of the nodes into hub and non-hub requires a method that goes beyond just degree. One method for this is the Borgatti-Everett (BE) core-periphery detection algorithm by Borgatti & Everett (2000). BE finds a partition of nodes into a dense core and sparse periphery by comparing the connectivity in the observed graph to an ideal core-periphery pattern, and assigns each node without the need for hyperparameters. We use the results of the algorithm as binary attributes.

**Designing a binary sensitive attribute for Chameleon.**  The Chameleon dataset is a Wikipedia page-page network, first introduced by Pei et al. (2020) and later refined by Platonov et al. (2023b), whose version we use. It is often used as a heterophilic benchmark, as, e.g., in Luo et al. (2024).

The Chameleon dataset has 5-ary sensitive attributes. To obtain binary sensitive attributes while keeping the heterophilic mixing, we try all possible binary partitions and select the binary partition that minimizes the fraction of within-group edges. To keep group sizes roughly balanced, we divide the groups into one group of two and one group of three, giving $\binom{5}{3} = 10$ possible groups $S$. To choose the most heterophilic group $S^\star$, we find the group with minimal absolute edge homophily $h_{\text{edge}}$ (Eq. 4), yielding $h_{\text{edge}}(S^\star) = 0.4955$. By Eq. 9, we get $h_{\text{edge}}^{\text{rand}} \approx 0.573$ as random baseline. Since $h_{\text{edge}}(S^\star) < h_{\text{edge}}^{\text{rand}}$, the found attribute distribution exhibits heterophily relative to chance, which we qualify as "Medium heterophily" in Table 10.

**Results of additional datasets.**  Table 12 shows the results of training on the dataset. MORAL predicts GARN with near-perfect scores, similar to the scores on other datasets described in Section 4.1.

MORAL performs even better on Chameleon. By comparing Chameleon's $h_{\text{edge}} \approx 0.5$ to results of synthetic homophily experiments with a similar absolute edge homophily, e.g. those in Figure 3, we see NDKL is

comparable between the synthetic and real-data setting, though Precision is in a significantly lower range in the synthetic setting. Most plausibly, this can be explained by the difference in features between the scenarios: the DPAH features are non-informative with Gaussian i.i.d. features ($d = 16$), whereas Chameleon has high-dimensional ($d = 2326$) real features, likely making prediction substantially easier.

Overall, the GARN results suggest that MORAL's fairness-utility behaviour generalizes to a different topology. On Chameleon, MORAL exhibits comparable exposure-based fairness behaviour in a heterophilous real-world setting, supporting transfer of the homophily stress-test conclusions beyond synthetic graphs.

Table 12: NDKL, AWRF, Precision, and NDCG at $K = 1000$ for the additional datasets.

| Model | Metric | GLOBAL AIRPORT ROUTES NETWORK | CHAMELEON |
|-------|--------|-------------------------------|-----------|
| **MORAL** | $\Delta_{DP}$@1000 | $0.0164 \pm 0.0015$ | $0.0011 \pm 0.0009$ |
| | NDKL@1000 | $0.0053 \pm 0.0000$ | $0.0074 \pm 0.0000$ |
| | AWRF@1000 | $0.0009 \pm 0.0000$ | $0.0015 \pm 0.0000$ |
| | Precision@1000 | $0.9587 \pm 0.0090$ | $0.9860 \pm 0.0065$ |
| | NDCG@1000 | $0.9967 \pm 0.0010$ | $0.9994 \pm 0.0003$ |

## G  Ablation: pre-rerank vs. post-rerank decoupled predictions

To isolate the contribution of MORAL's greedy KL aggregation, we compare fairness metrics before and after reranking while keeping the same decoupled subgroup-pair predictors fixed. Table 13 shows this comparison.

Table 13: Fairness before and after MORAL's greedy KL reranking, holding the same decoupled subgroup-pair predictors fixed. Pre-rerank denotes the score-based ranking produced directly by the decoupled predictors; post-rerank denotes the final MORAL ranking after greedy KL-aggregation. $\Delta$ = post-rerank $-$ pre-rerank, so negative values indicate fairness improvement. Values taken at $K = 1000$.

| Metric | Type | CREDIT | FACEBOOK | GERMAN | NBA | POKEC-N | POKEC-Z | CHAMELEON | GARN |
|--------|------|--------|----------|--------|-----|---------|---------|-----------|------|
| NDKL | pre | $0.0090 \pm 0.0027$ | $0.0277 \pm 0.0037$ | $0.0131 \pm 0.0027$ | $0.0210 \pm 0.0061$ | $0.0215 \pm 0.0012$ | $0.0213 \pm 0.0040$ | $0.0214 \pm 0.0008$ | $0.0222 \pm 0.0121$ |
| | post | $0.0038 \pm 0.0000$ | $0.0084 \pm 0.0000$ | $0.0068 \pm 0.0000$ | $0.0059 \pm 0.0000$ | $0.0073 \pm 0.0000$ | $0.0071 \pm 0.0000$ | $0.0074 \pm 0.0000$ | $0.0053 \pm 0.0000$ |
| | $\Delta$ | $(-0.0052)$ | $(-0.0193)$ | $(-0.0063)$ | $(-0.0151)$ | $(-0.0142)$ | $(-0.0142)$ | $(-0.0140)$ | $(-0.0169)$ |
| AWRF | pre | $0.0194 \pm 0.0102$ | $0.0325 \pm 0.0196$ | $0.0378 \pm 0.0094$ | $0.0207 \pm 0.0047$ | $0.0323 \pm 0.0061$ | $0.0261 \pm 0.0106$ | $0.0197 \pm 0.0087$ | $0.0196 \pm 0.0069$ |
| | post | $0.0021 \pm 0.0000$ | $0.0032 \pm 0.0000$ | $0.0021 \pm 0.0000$ | $0.0020 \pm 0.0000$ | $0.0023 \pm 0.0000$ | $0.0012 \pm 0.0000$ | $0.0015 \pm 0.0000$ | $0.0009 \pm 0.0000$ |
| | $\Delta$ | $(-0.0173)$ | $(-0.0293)$ | $(-0.0357)$ | $(-0.0187)$ | $(-0.0300)$ | $(-0.0249)$ | $(-0.0182)$ | $(-0.0187)$ |
| $\Delta_{DP}$ | pre | $0.0474 \pm 0.0026$ | $0.0074 \pm 0.0041$ | $0.0134 \pm 0.0043$ | $0.0009 \pm 0.0007$ | $0.0005 \pm 0.0003$ | $0.0131 \pm 0.0009$ | $0.0088 \pm 0.0065$ | $0.0339 \pm 0.0057$ |
| | post | $0.0000 \pm 0.0000$ | $0.0004 \pm 0.0001$ | $0.0110 \pm 0.0014$ | $0.0173 \pm 0.0055$ | $0.0000 \pm 0.0000$ | $0.0000 \pm 0.0000$ | $0.0011 \pm 0.0009$ | $0.0164 \pm 0.0015$ |
| | $\Delta$ | $(-0.0474)$ | $(-0.0070)$ | $(-0.0024)$ | $(+0.0164)$ | $(-0.0005)$ | $(-0.0131)$ | $(-0.0077)$ | $(-0.0175)$ |

Across all datasets, greedy KL-reranking substantially improves the two exposure-based fairness metrics relative to the same decoupled predictors before reranking, while $\Delta_{DP}$ is much less aligned with these changes: it improves on 6/7 datasets, but worsens on NBA. Averaged across datasets, NDKL decreases from 0.0197 to 0.0065, a relative reduction of 66.9%. AWRF decreases from 0.0260 to 0.0019, a relative reduction of 92.6%. This indicates that a substantial part of MORAL's fairness gains comes from the reranking step itself (Claim **C3**) as measured by NDKL (Claim **C2**), though undetected by $\Delta_{DP}$ (Claim **C1**).

# H    Results from the original paper

In this Appendix, we include the main results from the original paper (Mattos et al., 2025) for reference. Figure 11 shows the proportions of edge types in the top-100 per method they used compared to the original graph distribution. Table 14 compares performance of all their approaches by Precision@1000 and NDKL.

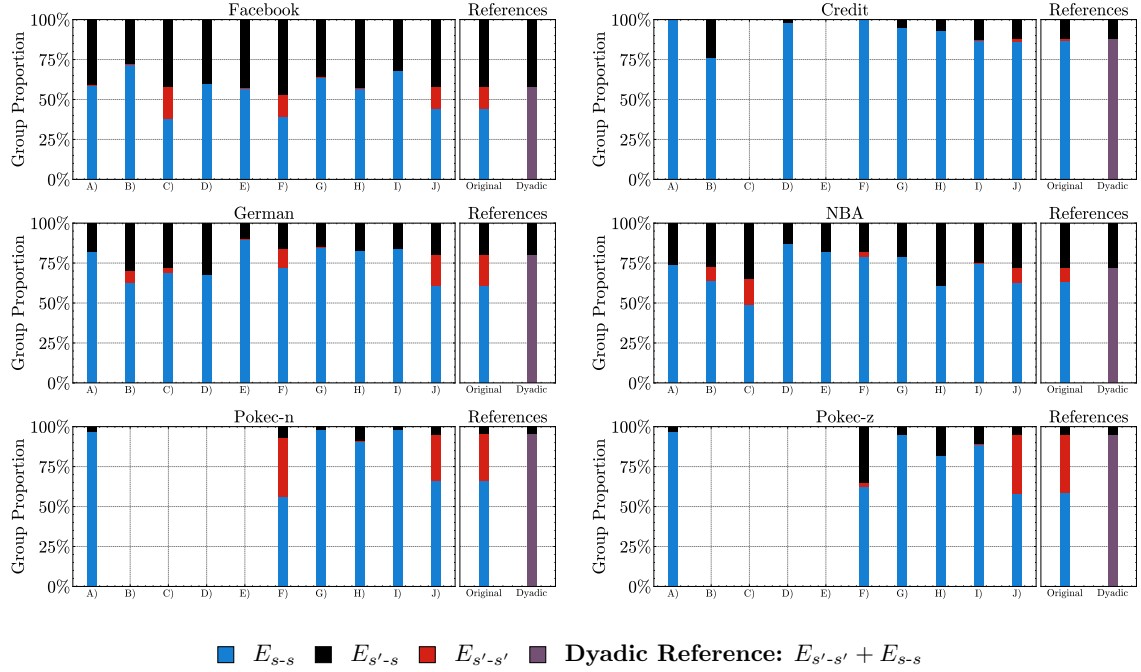

A) UGE  B) EDITS  C) FairAdj  D) FairEGM  E) FairLP  F) GRAPHAIR  G) FairWalk  H) DELTR  I) DetConstSort  J) MORAL

Figure 11: Adopted from Mattos et al. (2025). Proportions of pair types in the top-100 predictions by method, compared against the original graph distribution and an optimal dyadic fairness reference. Colors: $E_{s\text{-}s}$ (**blue**), $E_{s'\text{-}s}$ (**black**), and $E_{s'\text{-}s'}$ (**red**). In the dyadic fairness reference, **purple** represents the combined proportion of $E_{s'\text{-}s'}$ and $E_{s\text{-}s}$ pairs. Missing bars indicate an OOM error.

Table 14: Adopted from Mattos et al. (2025). Fairness performance comparison of all approaches considered ($k = 1000$). Lower NDKL and higher Precision@1000 are better. Best NDKL and Precision@1000 values are in **bold** and underline, respectively.

| Method | | Facebook | Credit | German | NBA | Pokec-n | Pokec-z |
|---|---|---|---|---|---|---|---|
| UGE | NDKL | $0.05 \pm 0.00$ | $0.80 \pm 0.07$ | $0.08 \pm 0.03$ | $0.07 \pm 0.02$ | $0.06 \pm 0.00$ | $0.06 \pm 0.00$ |
| | Precision@1000 | $0.97 \pm 0.00$ | $1.00 \pm 0.00$ | $0.69 \pm 0.01$ | $0.58 \pm 0.01$ | $0.90 \pm 0.04$ | $0.91 \pm 0.04$ |
| EDITS | NDKL | $0.21 \pm 0.07$ | $0.04 \pm 0.02$ | $0.08 \pm 0.02$ | $0.08 \pm 0.01$ | OOM | OOM |
| | Precision@1000 | $0.96 \pm 0.00$ | $0.36 \pm 0.17$ | $0.42 \pm 0.20$ | $0.49 \pm 0.00$ | OOM | OOM |
| GRAPHAIR | NDKL | $0.13 \pm 0.03$ | $0.67 \pm 0.22$ | $0.07 \pm 0.02$ | $0.09 \pm 0.01$ | $0.09 \pm 0.03$ | $0.26 \pm 0.25$ |
| | Precision@1000 | $0.96 \pm 0.01$ | $1.00 \pm 0.00$ | $0.73 \pm 0.01$ | $0.69 \pm 0.01$ | $0.97 \pm 0.03$ | $1.00 \pm 0.00$ |
| FairEGM | NDKL | $0.09 \pm 0.01$ | $0.11 \pm 0.00$ | $0.05 \pm 0.01$ | $0.07 \pm 0.01$ | OOM | OOM |
| | Precision@1000 | $0.97 \pm 0.00$ | $1.00 \pm 0.00$ | $0.62 \pm 0.00$ | $0.60 \pm 0.01$ | OOM | OOM |
| FairLP | NDKL | $0.18 \pm 0.00$ | OOM | $0.06 \pm 0.00$ | $0.20 \pm 0.00$ | OOM | OOM |
| | Precision@1000 | $0.99 \pm 0.00$ | OOM | $0.97 \pm 0.00$ | $0.86 \pm 0.00$ | OOM | OOM |
| FairWalk | NDKL | $0.06 \pm 0.01$ | $0.06 \pm 0.03$ | $0.11 \pm 0.02$ | $0.06 \pm 0.01$ | $0.07 \pm 0.01$ | $0.07 \pm 0.00$ |
| | Precision@1000 | $0.96 \pm 0.00$ | $1.00 \pm 0.00$ | $0.94 \pm 0.00$ | $0.55 \pm 0.01$ | $1.00 \pm 0.00$ | $1.00 \pm 0.00$ |
| FairAdj | NDKL | $0.10 \pm 0.05$ | OOM | $0.10 \pm 0.01$ | $0.11 \pm 0.05$ | OOM | OOM |
| | Precision@1000 | $0.42 \pm 0.01$ | OOM | $0.54 \pm 0.01$ | $0.50 \pm 0.01$ | OOM | OOM |
| DetConstSort | NDKL | $0.15 \pm 0.00$ | $0.06 \pm 0.00$ | $0.04 \pm 0.00$ | $0.09 \pm 0.00$ | $0.07 \pm 0.00$ | $0.23 \pm 0.00$ |
| | Precision@1000 | $0.00 \pm 0.00$ | $0.00 \pm 0.00$ | $0.55 \pm 0.00$ | $0.21 \pm 0.00$ | $0.07 \pm 0.00$ | $0.01 \pm 0.00$ |
| DELTR | NDKL | $0.10 \pm 0.03$ | $0.03 \pm 0.00$ | $0.09 \pm 0.06$ | $0.09 \pm 0.02$ | $0.23 \pm 0.23$ | $0.22 \pm 0.20$ |
| | Precision@1000 | $0.91 \pm 0.05$ | $0.56 \pm 0.29$ | $0.31 \pm 0.44$ | $0.43 \pm 0.24$ | $0.65 \pm 0.01$ | $0.48 \pm 0.28$ |
| MORAL | NDKL | $\mathbf{0.04 \pm 0.00}$ | $\mathbf{0.01 \pm 0.00}$ | $\mathbf{0.03 \pm 0.00}$ | $\mathbf{0.02 \pm 0.00}$ | $\mathbf{0.03 \pm 0.00}$ | $\mathbf{0.04 \pm 0.00}$ |
| | Precision@1000 | $0.95 \pm 0.01$ | $1.00 \pm 0.00$ | $0.96 \pm 0.00$ | $0.80 \pm 0.00$ | $0.98 \pm 0.00$ | $0.98 \pm 0.00$ |

