# OpenReview forum: "Fairness in Link Prediction Beyond Demographic Parity: A Reproducibility Study"
_TMLR — Accepted by TMLR_

### Review · Reviewer_TDwt · 2026-03-23

**Summary Of Contributions:**

Recent literature introduced the notion of *dyadic* fairness in link predictions by measuring the disparity between intra- and inter-group predictions. Very recently, Mattos et al. (AAAI 2026) identified limitations of this definition, introduced a rank-aware, exposure-based metric termed Normalized Discounted KL-divergence (NDKL), and proposed a post-processing algorithm termed MORAL that debiases predictions.

The submitted work reproduces and confirms the findings of Mattos et al., while identifying several inaccuracies in the implementation associated with the original paper. The authors also extend the prior work by introducing a new stress-test, prodiving a robustness analysis of MORAL using different fairness and utility metrics, and generalizing to categorical (i.e., not necessarily binary) sensitive attributes.

**Audience:**

No

**Audience Explanation:**

This submission is a bit borderline on this criterion:
- The authors identify flaws / discrepancies in the implementation of the recent work of Mattos et al. This is particularly relevant to the authors of the original paper as well as researchers who would follow up using their code.
- That said, given that the main claims of Mattos et al. are successfully validated and extended to other settings and metrics, I personally found the submission not as interesting as reproduction results that either overturn flawed findings or identify limitations of existing methods in other settings.

**Broader Impact Concerns:**

None.

**Claims And Evidence:**

Yes

**Claims Explanation:**

The methodology of the reproduction and extensions seems valid and rigorous to me. My only complaint was on the presentation being a bit unclear at times (see "Requested Changes" below), but this should be fixable during revision.

**Requested Changes:**

Minor comments:
- Bottom of Page 2: $n$ should be defined when it first appears (or replaced by $\mathbb{V}$).
- Page 3, Equation (2): This definition is unclear about the underlying distribution of the pair $(u, v)$ (e.g., is it drawn uniformly from $\mathbb{C}$?) and how the prediction $\hat y$ relates to $(u, v)$.
- Page 3, Properties 1 and 2: It is unclear at this time what $\mathbf{\pi}$ stands for.
- Page 3, Equation (3): What is the subscript $i$ in $\delta_i$?
- Bottom of Page 8: There is a sentence that starts with ")zero".

---

> ### Author Response · Authors · 2026-04-08
> **Response to Reviewer TDwt (Part 1/2)**
>
> Thank you for bringing these notation and definition issues to our attention. We have revised the manuscript accordingly, as detailed below.
> > Bottom of Page 2: $n$ should be defined when it first appears (or replaced by $\mathbb{V}$).
>
> At the bottom of page 2, in the definition of the sensitive attribute vector $\mathbf{s} \in \lbrace 0,1 \rbrace^{|\mathbb{V}|}$ (previously  $\mathbf{s} \in \lbrace 0,1 \rbrace ^{n}$), we changed the undefined $n$ to $|\mathbb{V}|$, to accurately denote that the sensitive attribute vector contains one binary attribute value for each node in the node set $\mathbb{V}$.
>
> > Page 3, Equation (2): This definition is unclear about the underlying distribution of the pair $(u,v)$ (e.g., is it drawn uniformly from $\mathbb{C}$?) and how the prediction $y$ relates to $(u ,v)$.
>
> We agree that the original demographic parity definition (Eq. (2)) was underspecified and too general for our ranking-based setting; thank you for pointing this out. The original definition stated that demographic parity measures the difference between the model's predicted positive-link probability for intra-group pairs and for inter-group pairs; it did not, however, specify how to actually calculate this in our setting.
>
> To revise the definition, make the quantity unambiguous, and to align it with the released code implementation, we now define dyadic demographic parity at cut-off $K$ as the empirical gap in average predicted link probability between the intra- and inter-group candidate pairs within the top-$K$:
>
> $$
> \Delta_{\mathrm{DP}} = |\frac{a}{b} - \frac{c}{d}|,
> $$
> where
> $$
> a = \sum_{(u,v) \in R_{1:K} {\cap} E_{\mathrm{intra}}} f(u,v),
> $$
> $$
> b = | R_{1:K} {\cap} E_{\mathrm{intra}} |,
> $$
> $$
> c = \sum_{(u,v) \in R_{1:K} {\cap} E_{\mathrm{inter}}} f(u,v),
> $$
> $$
> d = | R_{1:K} {\cap} E_{\mathrm{inter}} |.
> $$
> (The Markdown/LaTeX parser and/or renderer did not seem to process the complete equation at once nor did it process blackboard bold notation (\mathbb{} in LaTeX) correctly. Hence, we split the equation up, introducing $a, b, c, d$, and use capital letters for sets, e.g. $E_\mathrm{intra}$, instead of blackboard bold.) In the definition, $R_{1:K}$ denotes the top-$K$ prefix of the ranking and $f(u,v) \in [0,1]$ the predicted link probability for candidate node pair $(u,v)$. This revised version makes the original notion, which was too general, more concrete: it measures the absolute difference between the average predicted link probability of intra-group candidate pairs and that of inter-group candidate pairs among the pairs appearing in the top-$K$ positions of the ranking. We apologize for the earlier underspecification and thank the reviewer for drawing our attention to it. This clarification is important, as demographic parity is one of the main fairness metrics in this work, and should be specified precisely.
>
> > Page 3, Properties 1 and 2: It is unclear at this time what $\pi$ stands for.
>
> The target distribution $\mathbf{\pi}$ and empirical distribution $\hat{\mathbf{\pi}}$ are now introduced where they are first used, as opposed to later in the text: $\boldsymbol{\pi}$ denotes a discrete probability distribution over $T$ subgroup-pair types, and
>
> $$\hat{\boldsymbol{\pi}}_k (\mathbf{R})$$
>
> the predicted distribution induced by the top-$k$ prefix of $\mathbf{R}$. In the binary case, this gives
>
> $$\boldsymbol{\pi} = {(\pi_{0\text{-}0},\pi_{0\text{-}1},\pi_{1\text{-}1})}$$
>
> and
>
> $$\hat{\boldsymbol{\pi}}_k = (\hat{\pi}_e,\hat{\pi}_f,\hat{\pi}_g), $$
>
> where
>
> $$e = k,0\text{-}0,$$
> $$f = k,0\text{-}1,$$
> $$g = k,1\text{-}1,$$
>
> introducing $e, f, g$ just here for the subscripts to correctly render. We hope this clarifies your concern.
>
> > Page 3, Equation (3): What is the subscript $i$ in $\delta_i$?
>
> In the NDKL definition, it was indeed unclear what the indexer $i$ in the normaliser denoted; we introduce a new variable
> $$Z = \sum_{i=1}^{K}\frac{1}{\log_{2}(i+1)},$$
>
> to clarify the NDKL’s normalisation, giving as NDKL definition
>
> $$\mathrm{NDKL} = \frac{1}{Z}\sum_{k=1}^{K} \delta_kD_{\mathrm{KL}}(\hat{\boldsymbol{\pi}}_k, \boldsymbol{\pi}).$$
>
> > Bottom of Page 8: There is a sentence that starts with ")zero".
>
> Thank you for this detailed comment! We adjusted the line skipping inconsistency causing the sentence at the bottom of page 8 to start with “)zero”. Furthermore, we put effort into making sure the rest of the script does not contain typesetting errors.

---

> ### Author Response · Authors · 2026-04-08
> **Response to Reviewer TDwt (Part 2/2)**
>
> We appreciate the point given on the relevance of the paper; it seems we have work to do in highlighting the importance of our contributions for the fair machine learning community.
>
> We'd like to emphasize why we believe this work is relevant for the TMLR audience. While our study ultimately supports the main claims of Mattos et al. [1], our contributions are three-fold: (1) correcting the original implementation and providing a publicly available, fully-reproducible, corrected implementation on GitHub; (2) confirming the main claims of the paper (demographic parity is limited for exposure-based fairness, NDKL is adequate for exposure-based fairness, and MORAL is effective for debiasing ranked outputs of decoupled link predictors); and (3) assessing the robustness of the main claims under diverse conditions. We briefly elaborate on these three points below.
>
> * Firstly, we had to correct multiple inconsistencies and missing components in the released implementation, for example: demographic parity was implemented using the sensitive attribute of only one endpoint rather than edge membership in the intra- or inter-group, and the NDKL implementation was missing altogether. Considering demographic parity and NDKL are the central fairness measures of the work, these are significant corrections. Furthermore, MORAL itself contained an implementation issue in the greedy KL-aggregation mechanism. Overall, we think these corrections help anyone who wants to use, compare against, or build on MORAL, especially because MORAL consistently performs best against the reproduced baselines.
> * Secondly, we believe a careful confirmation of the main claims is itself valuable for the fair machine learning community, especially because the claims concern fairness measurement in a graph-learning setting that still has many open methodological questions, for example around evaluation [2, 3], but also because the claims concern fairness intervention, where even small improvements in exposure (re)distribution can have large effects on downstream consequences of exposure bias [4]. We do not overturn the original conclusions, but confirm they are robust enough to build on.
> * Thirdly, the extensions go beyond the original setting and test whether the claims remain robust under diverse circumstances and stresses. (1) By evaluating MORAL on a broad, general range of homophily scenarios (which span realistic homophily scenarios of, e.g., social graphs), we show the method works and also in which situations it becomes most challenging. This is useful for practitioners to understand when fairness-aware re-ranking is likely to be most difficult or impactful. (2) The metric robustness analysis introduces a subgroup-pair adapted AWRF, which, on the one hand, confirms the effective mitigation of exposure bias next to the NDKL, and, on the other hand, connects the work with the information retrieval literature, where the AWRF is a commonly used exposure-based fairness measure (see, e.g., [5] or [6]). (3) The extension to higher-cardinality sensitive attributes is important because many realistic sensitive attributes are non-binary, making the evaluation more realistic, while also exposing an important practical limitation: as cardinality increases, fairness-aware exposure redistribution becomes harder and more computationally demanding. We believe this is a valuable finding, since it identifies a concrete research direction for future work.
>
> We have tried to reflect this importance in the text more prominently, including an updated motivation for the extensions in Section 3.6.
>
> [1] Mattos, J., Lina, D. H., & Silva, A. (2026, March). Breaking the dyadic barrier: Rethinking fairness in link prediction beyond demographic parity. In Proceedings of the AAAI Conference on Artificial Intelligence (Vol. 40, No. 29, pp. 24335-24343).
>
> [2] Laclau, C., Largeron, C., & Choudhary, M. (2022). A survey on fairness for machine learning on graphs. arXiv preprint arXiv:2205.05396.
>
> [3] Chen, A., Rossi, R. A., Park, N., Trivedi, P., Wang, Y., Yu, T., ... & Ahmed, N. K. (2024). Fairness-aware graph neural networks: A survey. ACM Transactions on Knowledge Discovery from Data, 18(6), 1-23.
>
> [4] Singh, A., & Joachims, T. (2018, July). Fairness of exposure in rankings. In Proceedings of the 24th ACM SIGKDD international conference on knowledge discovery & data mining (pp. 2219-2228).
>
> [5] Abolghasemi, A., Azzopardi, L., Askari, A., de Rijke, M., & Verberne, S. (2024, March). Measuring bias in a ranked list using term-based representations. In European Conference on Information Retrieval (pp. 3-19). Cham: Springer Nature Switzerland.
>
> [6] Ekstrand, M. D., McDonald, G., Raj, A., & Johnson, I. (2023). Overview of the TREC 2022 fair ranking track. arXiv preprint arXiv:2302.05558.

---

### Review · Reviewer_m7nB · 2026-03-24

**Summary Of Contributions:**

This paper reproduces the experiments of Mattos et al (2025) about fairness in link prediction on graphs. In particular, it reimplements their code, correcting some points that were not made clear in the original paper and everything is accessible on a dedicated github page.

Beyond reproducing the experiments of Mattos et al (2025), this work proposes a few additional experiments, testing on synthetic data the robustness of the MORAL algorithm to i) asymmetric homophily, ii) choice of metric, iii) non-binary sensitive attributes.

**Additional Comments:**

While the github page has a clear interest from a reproducibility point of view, my global opinion is that the current paper needs a clear improvement in writing and introducing the setting and experimental setup.

**Audience:**

Yes

**Audience Explanation:**

TMLR editorial policy invites authors to submit papers that contain reproducibility studies of previously published results or claims. The work of Mattos et al. has been recently published at AAAI and lies in the algorithmic fairness field, which I believe is of interest to some TMLR's audience.

As a note though, this paper is very recent and only has been cited once according to Google Scholar. So it is yet hard to assess whether such a work deserves enough attention for a reproducibility work.

**Claims And Evidence:**

Yes

**Claims Explanation:**

The claims mainly derive from the proposed code, which is available on a public dedicated github.

**Requested Changes:**

I am not an expert of the domain, so I will only give educated guess.

My main concern is that, being a reproducibility work, I believe this paper should help the reader understand the experimental setup of Mattos et al. (2025) and how to implement/run such experiments on the reader's own.

However, I find the current writing of the paper very unclear/confusing/incomplete. Maybe it is easier to read for an expert, but I was somehow already lost after just reading the abstract. To me, the considered problem is insufficiently introduced. I believe the goal of such a paper is also to help non-experts understanding the problematic considered here, but it is clearly not the case from my point of view.

Maybe this is due to my theoretician background, but many concepts are either not defined, or not done clearly. I would have appreciated more rigorous definitions, which I believe are important in fairness, where very subtle changes (eg in probabilty conditioning) have huge impact on the evaluated criterion.

Moreover, I may be too rigid, but the used notations are a mathematical crime from my point of view: using $\mathbb{C}$ and $\mathbb{E}$ for sets, while it is now standard that they are used for the complex set and the expectation. For the latter, it is even more confusing, as fairness typically involves probabilities and expectations. Moreover, this depart from traditional graph notations, using $V$ and $E$ respectively for vertices and edges.

I will not give a complete list of all the parts I find confusing, but as an illustration, even the key quantity of the paper $\Delta_{text{DP}}$ is not clear from my point of view:
  - what is $p$ here? I guess some probability, but over what?
  - is $\hat{y}$ a fixed value here? If I have to guess, I would say no, and that the absolute value in equation (2) should actually be a total variation, but note this is totally not clear for the reader

As another example, it is not clearly stated what the distributions $\pi$ and $\hat{\pi}$ stand for. Because of that, it is not even clear what the NDKL stands for. Plus, what is the $\frac{1}{\delta_i}$ in Equation (3)? i is fixed?

---

> ### Author Response · Authors · 2026-04-12
> **Response to Reviewer m7nB**
>
> Thank you for this review and for sharing your perspective as a non-specialist reader. This was very valuable to us because it highlighted where the paper assumed too much familiarity. We revised the mathematical definitions, notation, and the exposition to make the paper more explicit and accessible.
>
> **Definitions.** Clarifications to definitions include:
> * Specification of the used target distribution $\mathbf{\pi}$ and empirical distribution $\hat{\mathbf{\pi}}$ where they are first used.
> * We clarified the NDKL normaliser (previously $\delta_{i}$) by introducing an additional variable $Z$:
> $$
> Z = \sum_{i=1}^{K}\frac{1}{\log_{2}(i+1)}.
> $$
> * Further specifying the definition for demographic parity ($\Delta_\mathrm{DP}$). Whereas the $\Delta_\mathrm{DP}$ definition by Mattos et al. (2025) was more of a general binary classifier definition (which assumed the existence of a positive prediction $\hat{Y}=1$ and a probability distribution $p(\cdot)$ over candidate pairs $(u,v)$ in the candidate ranking, as the review points out), our new definition is aligned with our experimental implementation. It defines dyadic demographic parity at cut-off $K$ as the empirical gap in average predicted link probability between the intra- and inter-group candidate pairs within the top-$K$. (We give the lengthy mathematical definition in the reply to Reviewer TDwt, placed on March 23rd.) In short, the original definition was indeed too general, and the new definition makes $\Delta_\mathrm{DP}$ more concrete. To emphasize the difference, we discuss the distinction between the original formula and our adaptation in a footnote.
>
> **Notation.** On notation, the review notes:
>
> > Moreover, I may be too rigid, but the used notations are a mathematical crime from my point of view: using $\mathbb{C}$ and $\mathbb{E}$ for sets, while it is now standard that they are used for the complex set and the expectation. For the latter, it is even more confusing, as fairness typically involves probabilities and expectations. Moreover, this depart from traditional graph notations, using $V$ and $E$ respectively for vertices and edges.
>
> The TMLR template (https://jmlr.org/tmlr/author-guide.html) prescribes to use blackboard bold notation for sets. We agree it's confusing how, for instance, $\mathbb{C}$, usually used for the complex set, now is introduced as the set of candidate node pairs. Nonetheless, blackboard bold notation for sets is used often in ML literature, for example in Deep Learning by Goodfellow (https://github.com/goodfeli/dlbook_notation/blob/master/notation_example.pdf). While we considered adjusting the notation, we ultimately decided to keep the notation as is with the purpose of consistency and to not depart from the template. If you think there's a better solution to this, we would be more than happy to hear your view!
>
> **Exposition.** We revised the writing to improve readability for non-expert readers. On accessibility and understandability of the text, the review notes:
>
> > I find the current writing of the paper very unclear/confusing/incomplete. Maybe it is easier to read for an expert, but I was somehow already lost after just reading the abstract. To me, the considered problem is insufficiently introduced.
>
> To further pin down exactly where our text could be improved in clarity, we asked three colleagues unfamiliar with this literature to feedback the manuscript. This was very helpful and led to, besides general polishing of the complete text, some specific additions/deletions that we believe better introduce the problem setting and improve the readability, while also balancing the use of jargon with simple language in a better way. We give some examples below:
> * We rephrased the Abstract to now have simpler language without losing substance.
> * In the introduction, we give more examples to illustrate the problem we’re trying to solve, one of which is: “For example, in homelessness support systems, data-driven tools are used to distribute scarce services (Wilde et al., 2021; Moon et al., 2025), and if clients with certain
> sensitive attributes are ranked lower, they may receive fewer referrals or slower access to support (Rice & Young, 2025).”
> * We explain the limitations of demographic parity more intuitively now.
> * We also rephrased passages that previously assumed familiarity with concepts such as subgroup-pair types, exposure bias, and GNN expressivity. In particular, the term “group” was somewhat overloaded, which is now clarified.
>
> In summary, we agree with the review’s main point on clarity, and we believe the revised version is now more accessible and easier to read.

---

### Review · Reviewer_Vyua · 2026-04-05

**Summary Of Contributions:**

This work is a careful reproducibility study of Mattos et al. (2025) on fairness in ranked link prediction. Specifically, the authors revisit 3 core claims: (1) that dyadic demographic parity can hide within-group exposure bias, (2) that NDKL is better suited to capture such bias, and (3) that MORAL offers a favorable fairness–utility trade-off.

With regard to the contributions, in general, the paper also goes beyond reproduction by adding homophily stress tests with the alternative fairness metric AWRF and additional utility metric (NDCG@1000), and an extension to categorical sensitive attributes. A strong practical contribution is that they identify several mismatches between the paper and implementation, including how the target distribution is computed, how demographic parity is measured, inconsistencies in the Pokec sensitive attribute, and an incorrect KL-based aggregation step. They also note that NDKL and the splitting procedure were missing or underspecified and had to be reimplemented. Hence, I think the paper provides a bag of useful observations for fairness for link prediction tasks.

Strengths

- First of all, in terms of the scope, I believe this is a thoughtful and technically engaged reproducibility study. The paper does more than simple reproduction but also provides many other analyses: identifies several substantive inconsistencies between the original implementation and formal definitions, including the target distribution, the demographic parity computation, the treatment of sensitive attributes in Pokec, and the KL-based aggregation step.

- In general, the overall writing is clear based on its scope.

- Empirically, the study supports the main qualitative claims that demographic parity can miss within-group exposure bias, that NDKL is sensitive to this effect, and that MORAL achieves strong fairness with limited utility loss on the tested benchmarks. The added robustness checks with AWRF, NDCG, homophily stress tests, and categorical attributes strengthen the paper further.

Weaknesses

As I am not an expert in this domain, most of the weaknesses are essentially technical questions I have when reading the paper.

1. My main reservation is that the robustness analysis still remains somewhat close to the method’s own design assumptions.Specifically, the normative status of the target distribution is under-discussed: if preserving the observed subgroup-pair distribution is itself problematic, then low NDKL or AWRF may not imply fair outcomes in a broader sense. Hence, I am also not fully convinced by the practicality at larger attribute cardinalities, since MORAL requires decoupled predictors per subgroup-pair type and the paper itself highlights the resulting compute and scaling concerns. Please correct me if I'm wrong on this.

2. Could the authors test a fairness criterion based on a substantively different target rather than a different distance or aggregation over the same target? I'm asking this since AWRF still uses the same target distribution and the same log-discounted exposure setup, not sure how independent is this check really?

3. What would happen if the observed graph distribution is already structurally (or semantically) biased? In that case, why should matching $\pi$ be considered fairness-improving rather than bias-preserving?

4. MORAL is framed as a post-processing approach, yet it requires separate subgroup-pair predictors before aggregation. How should readers think about the true source of gains: the decoupled predictors, the greedy KL aggregation, or the combination? A more detailed ablation would help.

5. Minor: The paper evaluates at K=1000 and notes that this may hide where in the ranking MORAL helps most. Would the main conclusions still hold across a range of K values, especially on smaller graphs like NBA where the top-K cutoff covers a larger fraction of candidates and utility appears more sensitive?

6. The homophily stress test is informative, but it is based on synthetic DPAH graphs with i.i.d. features that do not encode group membership. How much should we expect the same non-monotonic fairness behavior around intermediate homophily to transfer to real graphs where features, degree effects, and group structure are entangled?

**Audience:**

Yes

**Audience Explanation:**

I think the work would be of interest to several communities, including fairness in machine learning, graph learning, and recommender systems.

**Claims And Evidence:**

Yes

**Claims Explanation:**

Yes, I checked that all claims can be supported by the empirical evaluation in this paper (though some can be further analyzed, see the questions above)

**Requested Changes:**

1. Clarify the main contribution and better connect the extensions into one coherent message (I would suggest putting this more explicitly as an individual section)

2. Better justify the choice of target distribution as the fairness objective, and discuss when it may be inappropriate (again correct me if I have any misunderstanding on this point)

3. Add stronger analysis to separate MORAL’s true gains from possible alignment with the evaluation metric (see the detailed questions above)

4. Include discussion or evidence on scalability as the number of sensitive groups grows.

---

> ### Author Response · Authors · 2026-04-13
> **Response to Reviewer Vyua (Part 1/3)**
>
> Thank you for your constructive review. We also appreciate your positive assessment of the paper’s scope, writing, empirical support, and practical value as a reproducibility study. We’ve used the suggestions and questions raised to sharpen the explanations throughout the text, and did some additional analyses to answer some of the questions in more detail.
> We will take overlapping topics between the 6 questions and 4 requested changes (**RC**s) and discuss them one-by-one.
>
>
> **Highlighting contributions.** Requested change 1 (**RC1**) notes:
>
> > Clarify the main contribution and better connect the extensions into one coherent message (I would suggest putting this more explicitly as an individual section)
>
> Thank you for pointing out we could clarify the contributions better. Of the changes to do this, the most notable one is adding a paragraph at the beginning of Section 3.6 of the methodology (where we explain each extension) to motivate the importance of every extension more explicitly and in one place.
>
>
> **Choice of target distribution.** Question 1, 2, 3 and **RC2** touch on the choice of target distribution, with **RC2**:
>
> > Better justify the choice of target distribution as the fairness objective, and discuss when it may be inappropriate (again correct me if I have any misunderstanding on this point)
>
> The concern about the validity of Property 1 (non-dyadic distribution-preserving fairness) and MORAL aiming for aligning with the original reference distribution $\boldsymbol{\pi}$ (which essentially boils down to Property 1 as well) is a very fair one. We think this cuts to the core of the problem of fairness, perhaps in a normative sense: what even is fair? We agree that we could have discussed this more, as this would be too important of a problem to let undiscussed in both the original text by Mattos et al. (2025) and our reproducibility endeavour. In line with your comments, we have extended the discussion on the choice of target in Section 3.3:
>
> “Notably, $\boldsymbol{\pi}$ is a design choice that, by choosing distribution-preserving fairness, aims to prevent algorithmic bias, rather than it being a universally correct definition of fairness (Mitchell et al., 2021). In this sense, because link predictors can inherit and amplify structural bias in ranked outputs (Gupta et al., 2021; Dai et al., 2024), Property 1 and MORAL's distribution-preserving target should be interpreted as a non-amplification objective, not a corrective one. In other words, if the original $\boldsymbol{\pi}$ reflects historical bias, then low NDKL just indicates agreement of the predicted $\hat{\boldsymbol{\pi}}$ with the original $\boldsymbol{\pi}$, not with fairness in a broader normative sense.”
>
> In addition, we’ve emphasized this limitation more clearly in the limitations part of the Discussion:
>
> “Finally, as mentioned in Section 3.3, preserving $\boldsymbol{\pi}$ is a design choice rather than a universally correct fairness target (Mitchell et al., 2021). Thus, the low NDKL and AWRF values achieved by MORAL should be interpreted as evidence of strong performance under distribution-preserving fairness, rather than for support of a more general claim about fairness, as the reference distribution $\boldsymbol{\pi}$ might contain biases in itself. Future work should compare fairness objectives that reflect different normative commitments.”
>
> So, to answer the questions in short: The choice of target distribution is one that aims to prevent further algorithmic amplification of existing biases, not one that is able to correct these existing biases.

---

> ### Author Response · Authors · 2026-04-13
> **Response to Reviewer Vyua (Part 2/3)**
>
> **Practical scalability limits.** Question 1 also goes into the practical scalability of MORAL, which requires decoupled predictors and hence scales unfavourably with the amount of subgroup-pair types. **RC4** touches on this too:
>
> > Include discussion or evidence on scalability as the number of sensitive groups grows.
>
> We agree that the scalability concern is real, and can be split into three dimensions: fairness, utility, and compute:
> * Concerning fairness and utility, we refined our interpretation of the higher-cardinality results and made this more explicit in the revised manuscript. For fairness, both NDKL and AWRF increase with the number of subgroup-pair types, although NDKL rises more steeply. We now emphasize more clearly that AWRF provides an important complementary view here, since the sharper increase in NDKL partly reflects its cumulative prefix-wise aggregation mechanism, rather than only stronger fairness degradation. For utility, we likewise now place more emphasis on NDCG than on Precision in the higher-cardinality setting. As subgroup cardinality grows, MORAL’s redistribution can shift more links across the cut-off, which affects Precision more because it is not position-weighted, whereas NDCG better reflects utility near the top of the ranking. Taken together, these results paint a somewhat more favourable picture of MORAL at higher cardinality than our earlier wording suggested: the redistribution problem does become harder, but the degradation is more moderate than we previously emphasized. We have revised the discussion (including the practical recommendations) accordingly to present this scalability concern more carefully and with more nuance.
> * Compute-wise, we agree that requiring decoupling raises a scalability concern as cardinality grows. The question of whether this requirement can be relaxed remains open and is, to our understanding after correspondence with Mattos et al., being explored by them in ongoing follow-up work, we do not attempt to resolve it here.
>
>
> **Ablation of the greedy KL reranking step.** **RC3** notes:
>
> > Add stronger analysis to separate MORAL’s true gains from possible alignment with the evaluation metric
>
> To address this requested change and question 4, we have added an ablation as Appendix G, where we compare the ranking obtained directly from the decoupled subgroup-pair predictors (pre-rerank) with the final MORAL ranking after greedy KL aggregation (post-rerank). This isolates the effect of the reranking step, not of the predictors themselves. (Indeed, this does not test the requirement of the decoupled predictors; to our understanding, this broader question is also being explored in ongoing follow-up work by the original authors, so we do not try to settle it here.) The results were actually quite interesting: the pre- vs. post-rerank comparison shows that greedy KL aggregation alone reduces mean NDKL by 66.9% and mean AWRF by 92.6%, showing that a substantial part of MORAL’s fairness gain comes from the reranking step itself.
>
> These results also speak directly to **RC3** itself (about possible metric alignment). We agree that, since MORAL is closely related to the fairness notion underlying NDKL, it is important to distinguish genuine gains from possible metric alignment. However, this alignment is not incidental: in our setting, fairness must be both subgroup-aware and rank-aware. Following Mattos et al., we view Property 1 (subgroup-aware, distribution-preserving fairness) and Property 2 (rank-awareness) as essential: we judge fairness of subgroups (Property 1; which demographic parity is invariant to) and we judge it using exposure (Property 2). Dropping either would amount to evaluating a different fairness notion outside the scope of this paper, rather than stress-testing the same one.
>
> For that reason, our robustness analysis keeps these properties fixed and instead varies the aggregation mechanism. Specifically, subgroup-pair-adapted AWRF remains subgroup-aware and exposure-based, yet differs fundamentally from NDKL in that it evaluates a single position-weighted exposure distribution rather than accumulating prefix-wise divergences. We therefore view AWRF as a meaningful challenge to MORAL within the paper’s intended fairness framework. The fact that MORAL continues to separate from the baselines under AWRF suggests that its improvements are not merely an artefact of optimizing NDKL’s exact formulation; in fact, the metric robustness analysis, the categorical-attribute extension, and the newly added ablation all point to AWRF possibly being a more stable measure of the fairness gains.
>
> We have revised the manuscript to make this distinction clearer, including in the ablation appendix, the metric robustness discussion, and the practical recommendations.

---

> ### Author Response · Authors · 2026-04-13
> **Response to Reviewer Vyua (Part 3/3)**
>
> **Cut-off analysis.** Question 5 notes:
>
> > Minor: The paper evaluates at K=1000 and notes that this may hide where in the ranking MORAL helps most. Would the main conclusions still hold across a range of K values, especially on smaller graphs like NBA where the top-K cutoff covers a larger fraction of candidates and utility appears more sensitive?
>
> We have added an analysis, “Sensitivity of MORAL to ranking cut-off $K$,” where we plot fairness and utility metrics as a function of $k$. To answer the question: the cut-off analysis suggests that MORAL’s fairness-utility patterns are broadly stable across cut-offs, though, indeed, on the NBA dataset (a relatively small graph) utility is more sensitive to the choice of cut-off. Fairness patterns remain stable across datasets, also NBA. We have added these lessons to the revised manuscript, including the new Appendix and the Discussion.
>
>
> **Transferability of synthetic homophily experiments to real-world data.** Question 6:
>
> > How much should we expect the same non-monotonic fairness behavior around intermediate homophily to transfer to real graphs where features, degree effects, and group structure are entangled?
>
> A discussion on to what degree behaviour or conclusions from the synthetic homophily experiments to real graph datasets was indeed missing in the main text (apart from a small note on it, hidden away in Appendix E); thank you for drawing our attention to this. To address this, we add notes in the Results and Discussion section, where we now discuss transferability separately for fairness and utility. Firstly, in the Results section, we introduce the notion of transferability by comparing the synthetic fairness and utility results with the real ones:
>
> “Comparing the synthetic heatmaps with Tables 2 and 3, NDKL appears to be of similar magnitude in both the synthetic and real-data setting. In contrast, Precision lies in a substantially lower range in the synthetic setting. Most plausibly, this can be explained by the difference in features between the scenarios: the DPAH features are non-informative with Gaussian i.i.d. features ($d = 16$), whereas the used datasets have real features of higher dimensionality (e.g., $d = 95$ for NBA), likely making prediction substantially easier.”
>
> Then, in the Discussion, we added that the transferability is likely hindered by the entanglement of the factors you mentioned: features, degree effects, and group structure. Still, since the NDKL and AWRF peaks respond consistently to absolute homophily caused by isolated mixing effects, we view this as evidence that absolute homophily has a structural effect on exposure disparities, rather than it being only an artefact of the synthetic generator. We have written this down more clearly in the text, where we have now separated (i) the transferability of synthetic to real results and (ii) the relevance of absolute homophily as a signal for exposure-based fairness.
>
>
> **Thank you!** Overall, we are grateful for these comments, which helped us improve both the scope and the presentation of the paper. We have revised the manuscript to make its assumptions clearer, its extensions more coherent, and its limitations more explicit. We hope the revised version addresses your concerns more satisfactorily, and we thank you again for the careful and constructive review.

---

### Author Response · Authors · 2026-04-08
**Revision in progress**

We are currently working on revising the manuscript based on all reviewer comments. We will upload the updated version later this week.

---

### Author Response · Authors · 2026-04-13
**Revised manuscript uploaded**

Thank you all for the thoughtful and constructive reviews. We have uploaded a revised manuscript that incorporates the changes discussed. Overall, the changes led to 1 additional page in the main text, and approximately 1.5 pages in the appendix. For ease of reference below, we denote the three reviews as **R1** (by reviewer TDwt), **R2** (by reviewer m7nB), and **R3** (by reviewer Vyua).

The most important parts of revision are:
* Clearer framing of the paper’s contributions and extensions (**R1**, **R3**), by, for example, motivating the importance of the extensions in a central place;
* More explicit and less ambiguous mathematical definitions (**R1**, **R2**), including an update of the demographic parity and NDKL definitions;
* Improved accessibility and exposition throughout the paper (mainly **R2**) by asking three colleagues unfamiliar with this literature to feedback the manuscript, leading to, for example, a revised abstract and introduction with reduced unnecessary jargon and more intuition and examples;
* Expanded discussion of the fairness objective and target distribution (**R3**) by emphasizing MORAL’s objective of distribution-preserving fairness (which closely aligns with Property 1 of the NDKL) should be interpreted as a non-amplification objective, not as a universally correct fairness target;
* New ablation isolating the effect of greedy KL reranking (**R3**): added an appendix that compares pre-rerank decoupled predictions with the final MORAL ranking after greedy KL aggregation;
* New cut-off analysis across ranking lengths (**R3**): added an appendix analysing fairness and utility as function of the variable cut-off $k$;
* More nuanced discussion of higher-cardinality scalability (**R3**); and
* Stronger discussion of transferability from synthetic homophily experiments to real graphs (**R3**), where we separate (i) the transferability of exact synthetic fairness-utility patterns and (ii) the relevance of absolute homophily as a structural signal for exposure-based fairness.

Overall, the revised manuscript is more explicit about its assumptions, clearer in its exposition, and stronger in the way it motivates the extensions and interprets their results. Thank you again for the feedback, we believe the paper is in a much better place because of it.

---

### Decision · Action_Editor_dYrS · 2026-05-12

**Recommendation:** Accept as is

**Audience:**

Yes

**Audience Explanation:**

Fairness and graph learning communities are likely interested in this paper, in particular their new correct implementation of the metrics.

**Claims And Evidence:**

Yes

**Claims Explanation:**

This paper is a reproducibility study of fairness in link prediction, assessing robustness. The main interesting component in the reproduction is that the paper fixes some issues in the prior experimentation. I think the fix is valuable and the reproduction of the paper is well executed. In addition, the paper exposes a number of additional tricks that are likely useful for practitioners.

Reviewers had concerns largely on the presentation, which was addressed by the authors in the rebuttal. Overall, they are all supportive of acceptance.